# TAFS: Task-aware Activation Function Search for Graph Neural Networks

## Abstract

Since the inception of Graph Neural Networks (GNNs), extensive research efforts have concentrated on enhancing graph convolution, refining pooling operations, devising robust training strategies, and advancing theoretical foundations. Notably, one critical facet of current GNN research remains conspicuously underexplored—the design of activation functions. Activation functions serve as pivotal components, imbuing GNNs with the essential capacity for non-linearity. Yet, the ubiquitous adoption of Rectified Linear Units (ReLU) persists. In our study, we embark on a mission to craft task-aware activation functions tailored for diverse GNN applications. We introduce TAFS (Task-aware Activation Function Search), an adept and efficient framework for activation function design. TAFS leverages a streamlined parameterization and frames the problem as a bi-level stochastic optimization challenge. To enhance the search for smooth activation functions, we incorporate additional Lipschitz regularization. Our approach automates the discovery of the optimal activation patterns, customizing them to suit any downstream task seamlessly. Crucially, this entire process unfolds end-to-end without imposing significant computational or memory overhead. Comprehensive experimentation underscores the efficacy of our method. We consistently achieve substantial improvements across a spectrum of tasks, including node classification over diverse graph data. Moreover, our approach surpasses state-of-the-art results in the realm of link-level tasks, particularly in biomedical applications.

## 1 Introduction

Graph Neural Networks (GNN) have demonstrated their prowess in modeling relationships within graph-structured data, as evidenced by their superior performance in various domains (Kipf & Welling, 2017; Velickovic et al., 2017; Hu et al., 2020; Xu et al., 2019). They have excelled in applications spanning biomedicine (Wu et al., 2023; Jiang et al., 2021), physical simulation (Sanchez-Gonzalez et al., 2020), material design (Reiser et al., 2022), sustainability (Donon et al., 2020), social network (Fan et al., 2019), transportation (Li et al., 2018b), recommendation (Wu et al., 2019), and more. Consequently, GNN models continue to captivate the attention of researchers across diverse scientific communities (Shi et al., 2020; Wang et al., 2022; Seo et al., 2020).

Despite the extensive body of literature, we must highlight a significant gap in current research, specifically the design of activation functions, a fundamental component used in nearly every GNN model. While Rectified Linear Unit (ReLU) (Nair & Hinton, 2010) is a prevalent choice for activation, it often falls short, as illustrated in Figure 1. Regrettably, GNN studies have hardly explored alternative activation functions. This oversight is critical, as the activation function plays a pivotal role in introducing non-linearity to GNNs. Without it, GNNs merely perform linear transformations on raw graph features.

In contrast, the Computer Vision community has spent decades exploring a wide array of manually

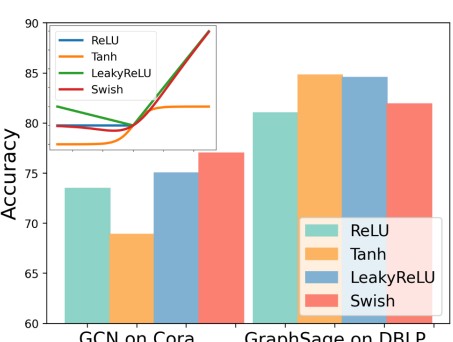

**Figure 1:** Activation function makes a big difference in GNN.

designed activation functions such as Sigmoid (LeCun et al., 1998), Tanh, ReLU (Nair & Hinton, 2010), and improved variants of ReLU (He et al., 2015; Clevert et al., 2016; Maas et al., 2013). However, transferring these manually crafted functions to different tasks poses challenges, and customizing new ones is a labor-intensive process. Furthermore, the marginal performance gains from human-designed functions diminish rapidly. To address this, researchers have proposed automated methods to discover tailored activation functions, which have demonstrated notable improvements in other network architectures like Convolutional Neural Networks (CNNs) or Recurrent Neural Networks (RNNs) (Ramachandran et al., 2018; Eger et al., 2018; Farzad et al., 2019).

Hence, our research question is *how can we design GNN activation functions to adapt effectively to various graph-based tasks, creating task-aware activation functions?*

Addressing this question poses two primary challenges. **Challenge #1:** Existing search algorithms are inefficient. Current activation function search methods suffer from over-parameterization and heavy computation. For example, APL (Agostinelli et al., 2015) introduces additional parameters for each neuron, which usually leads to at least ten times more parameters upon any model, thus significantly increasing the model complexity. Swish (Ramachandran et al., 2018) requires training a full network until convergence for each iteration, making it computationally burdensome. These issues render current search algorithms inefficient and less effective. **Challenge #2:** Current search methods lack support for non-differentiable objectives. GNN methods have wide applications regarding tasks of different levels (node, link, graph), many of which are evaluated by non-differentiable metrics. In the case of drug interaction prediction, we would like to know a certain positive (synergy) or negative (confliction) interaction exist. In fact, most drug pairs does not have positive or negative interactions. Receiver Operating Characteristic curve (ROC) which is not differentiable, is what we should use instead of accuracy. Similar application cases can be found in hit ratio of recommendation, latency optimization, hardware resources constraint, etc. Supporting these non-differentiable objectives would broaden the applicability of activation function search in diverse GNN tasks.

In this study, we embark on a systematic exploration of GNN activation function search—a first of its kind. We frame this search as a bi-level optimization problem, with the inner level optimizing GNN parameters and the outer level optimizing activation function parameters. We propose an efficient search algorithm that navigates through a compact search space. This space is characterized by universal approximators with additional smoothness constraints, facilitating the rapid discovery of high-quality functions, thereby addressing Challenge #1. Additionally, we tackle Challenge #2 by jointly considering non-differentiable objectives and potential activation function constraints. We incorporate these elements into a stochastic relaxation of the outer level optimization, removing the need to compute gradients for non-differentiable metrics used in GNN tasks. Our algorithm undergoes extensive experimentation across various GNN models, datasets, and objectives, consistently outperforming existing activation functions. By overcoming Challenges #1 and #2, our algorithm achieves *task-awareness* in GNN activation function design.

Our contributions can be summarized in three key points:

1. To the best of our knowledge, we are the first to propose activation function search in the context of Graph Neural Networks. Our work serves as a catalyst, drawing attention to this critical aspect of GNN model design and paving the way for future investigations.

2. We propose TAFS (**T**ask-aware **A**ctivation **F**unction **S**earch), a probabilistic search algorithm capable of efficiently exploring a regularized functional space to discover novel activation functions tailored for diverse downstream tasks.

3. Through comprehensive evaluations spanning node and link level tasks, we demonstrate that our algorithm enhances activation function design without requiring extensive manual effort and excels in optimizing non-differentiable objectives. We also conduct ablation studies to examine the searched activation functions, the impact of design choices, and algorithm efficiency.

## 2 RELATED WORKS

**Graph Neural Networks.** GNN is a power model in capturing relational information including typically GCN (Kipf & Welling, 2017), GAT (Velickovic et al., 2017), GIN (Xu et al., 2019), etc. Mathematically, in the context of GNN with a given network $\mathcal{G} = \{\mathcal{V}, \mathcal{E}\}$ containing node set $\mathcal{V}$ and edge set $\mathcal{E}$, the problem is formulated below:

$$z_u^{(l+1)} = \text{UPDATE}^{(l+1)} \left( h_u^{(l)}, \text{AGGREGATE}^{(l)} \left( \{ h_u^{(l)}, \forall v \in \mathcal{N}(u) \} \right) \right), \tag{1}$$

where $h_u^{(l)}$ is the latent representation of node $u$ at layer $l$ and $z_u$ is the pre-activation of node $u$. AGGREGATE (abbr. Agg) and UPDATE (abbr. Up) are core modules of GNN, denoting the different message passing operations used across the model for collecting and updating representations. The latent representation (respectively pre-activation) for all nodes constitute $\mathbf{H}$ (resp. $\mathbf{Z}$) and then we have the activation transformation: $\mathbf{H}^{(l+1)} = \sigma^{(l+1)}(\mathbf{Z}^{(l+1)})$, where $\mathbf{Z}$ is activated by function $\sigma$ at the $(l+1)$th layer.

Numerous studies have been devoted to ameliorate it through different aspects. For example, GCN (Kipf & Welling, 2017) simplifies graph convolution and derives performant GCN networks. GAT (Velickovic et al., 2017) proposes graph attention as a replacement to model global features. GNN-Pretrain (Hu et al., 2020) studies pretraining strategies at the node level and graph level to make GNN model work for transferable tasks. GIN (Xu et al., 2019) understands the fundamental question of graph expressiveness by discriminating Weisfeiler-Lehman graph isomorphism. GNN Co-training (Li et al., 2018a) connects Laplacian smoothing with graph convolution and studies the problem of oversmoothing. *However*, almost every GNN model uses ReLU as the activation function (Kipf & Welling, 2017; Velickovic et al., 2017; Xu et al., 2018; Huang et al., 2020; KC et al., 2022; Xu et al., 2019), leaving GNN activation function a missing research piece.

**Activation Function Design.** Since the early application of Sigmoid in Le-Net, activation functions have been considered as an important component until today (Hayou et al., 2019). In 2012, ReLU (Nair & Hinton, 2010) was proposed to train Boltzman Machines and soon extensively adopted in every neural network models. The study of activation function design happens mostly in CNN community, where a couple of milestone works include Swish (Ramachandran et al., 2018; Eger et al., 2018) and APL (Agostinelli et al., 2015). Swish proposes a Reinforcement Learning (RL)-based search algorithm to find appropriate activation functions in a discrete space. APL (Adaptive Piecewise Linear) uses linear hinge functions to approximate target patterns in a differentiable way. Comprehensive surveys of manual designed and parametric activation functions can be found in (Apicella et al., 2021; Dubey et al., 2022). Another notable work related to our research question is GReLU (Zhang et al., 2022), which tries to make GNN activation function adaptive by including graph convolution into the activation function. However, such design is not a typical univariate activation function. As a result, *no work yet has proposed novel activation functions designed under the context of GNN*.

## 3 PROBLEM FORMULATION AND CHALLENGES

Our research problem requires to propose a systematic way of designing adaptive activation functions that can be effectively integrated into GNN for downstream applications. Similar to Neural Architecture Search (NAS) (Liu et al., 2019), the activation function design could be modeled into a **bi-level optimization** problem:

$$\min_\alpha \mathcal{M}(w^*(\alpha), \alpha; \mathcal{D}_{\text{val}}) \quad \text{s.t.} \quad w^*(\alpha) = \arg\min_w \mathcal{L}(w, \alpha; \mathcal{D}_{\text{train}}), \tag{2}$$

where the inner level optimization learns $w$ weight of GNN and the outer level optimization learns $\alpha$ weights of activation function. Both levels may have different objective metrics $\mathcal{M}, \mathcal{L}$ depending on downstream applications.

Previous activation function search methods suffer from low efficiency (Challenge #1) and poor support of non-differentiable metrics (Challenge #2). On one hand, the efficiency bottleneck lies in the search space choice and search strategy design. The search space is crucial for search efficiency and requires careful consideration. The space should be proper both in candidate function number and effectiveness, making it a trade-off between quantity and quality. Then, the search strategy should be able to discover as quickly as possible the most suitable function candidate in the space. On the other hand, diverse GNN applications requires that the algorithm is able to tackle *any* downstream target metric, whether or not differentiable.

All these issues prevent us from using off-the-shelf algorithms for GNN activation function search. To this end, we need a novel parameterization of the search space, that is jointly designed with search strategy to allow efficient search, and we need to deal with differentiable and non-differentiable target metrics at the same time to enable general applications in all kinds of GNN tasks.

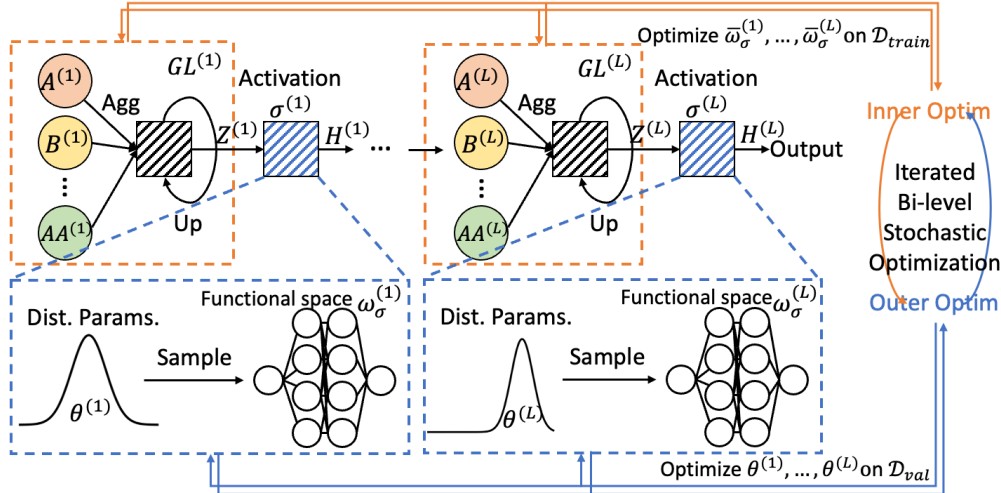

**Figure 2:** Algorithm framework. We replace the activation function with sampled weights from Gaussian distributions. The inner level optimization is the learning of distribution parameter $\theta^{(1)}, \ldots, \theta^{(L)}$ and the outer level optimization is the learning of GNN weights. Both levels are iterated for faster convergence.

# 4 THE PROPOSED METHOD

## 4.1 ALGORITHM FRAMEWORK

As illustrated in Figure 2 and Table 1, we propose TAFS to solve both challenges in a unified way. We follow the bi-level optimization formulation and search activation function represented by learnable parameters. Specifically, a typical GNN network of L layers can be represented as below:

$$\sigma^{(L)} \circ \text{GL}^{(L)} \cdots \sigma^{(2)} \circ \text{GL}^{(2)} \circ \sigma^{(1)} \circ \text{GL}^{(1)}(\mathbf{X}), \tag{3}$$

where Graph Layer (GL) denotes all the AGGREGATE (Agg) and UPDATE (Up) operations related to graph, $\mathbf{X}$ is the initial graph features. Note that different layers could use *different* activation functions whereas current GNN models tend to fix the same ReLU for every layer.

Denote by $w_\sigma$ all the parameters of activation functions, i.e. $\sigma^{(1)}, \ldots, \sigma^{(L)}$, and denote by $\overline{w}_\sigma$ all the parameters of GNN, i.e., parameters of $\text{GL}^{(1)}, \ldots, \text{GL}^{(L)}$. We propose a continuous implicit functional space to parameterize $w_\sigma$. This search space is expressive yet compact, with smoothness regularization induced by human prior. The parameter update process is stochastic to deal with *any* downstream objective especially non-differentiable metrics. The search algorithm is bi-level and end-to-end trained. The optimization step of the outer level (learning $w_\sigma$) and the inner level optimization (learning $\overline{w}_\sigma$) are iterated. In the following parts, we explain in sequence the design of search space, stochastic relaxation and search algorithm.

## 4.2 IMPLICIT FUNCTIONAL SEARCH SPACE

In order to facilitate activation function search, we propose a continuous implicit functional space that parameterizes the search space by universal approximators. This implicit functional space could be implemented by Multi-Layer Perception (MLP) to approximate target function. As in Figure 2, activation function parameters is equivalent to the parameters of MLP, denoted by $w_\sigma$.

It's worth noting that we employ MLP as a representative example of universal approximators, chosen for its simplicity, while retaining generality. However, it's crucial to emphasize that alternative implementations, such as Gaussian Mixtures or Radial Basis Functions (RBF), are entirely feasible.

In addition, we focus on the smooth functions such that the searched activation functions will not change dramatically if the pre-activation value $Z$ is slightly perturbed. Smooth functions are bounded by Lipschitz constant $c$, i.e. $|f(x) - f(y)| \leq c|x - y|$. As a result, the functional search space is

regularized by smoothness constraint $\{w_\sigma | w_\sigma \in R^{|w_\sigma|}, c < \gamma\}$, where $c$ is the Lipschitz constant of the function parameterized by $w_\sigma$ and $\gamma$ is the hard limit on $c$. Here, we model the constraint on Lipschitz constant as a regularization term denoted by $R(w_\sigma)$. Many references design Lipschitz constant as additional soft metric to be trained together with any loss (Hoffman et al., 2019; Weng et al., 2018; Liu et al., 2022). We use Jacobian regularization (Hoffman et al., 2019) without loss of generality $R(w_\sigma) = ||J(x)||_F = \{\Sigma_{i,j}[J_{i,j}(x)]^2\}$, where $J_{i,j}(x) = \frac{\partial h_i}{\partial x_j}(x)$ is the Jacobian matrix.

This design of our functional space encourages the discovery of smooth functions characterized by small Lipschitz constants. Notably, this characteristic aligns with existing manually designed functions, such as ReLU, Tanh, Sigmoid, and Swish, all of which exhibit 1-Lipschitz properties.

## 4.3 STOCHASTIC RELAXATION

GNN applications are diverse that in many cases, the preferred evaluation metrics may not be differentiable. As mentioned in Table 1 and related works, APL could not deal with non-differentiable metrics but Swish could due to the RL-based search algorithm. However, the search efficiency of Swish is far from satisfactory.

We propose to use stochastic relaxation that re-parameterizes the search space ($w_\sigma$) with a Gaussian distribution $p_{\theta_\sigma}(w_\sigma)$. The Gaussian distribution has its own parameters $\theta_\sigma$ and we sample the parameters of activation function $w_\sigma$ from the probability $p_{\theta_\sigma}(w_\sigma)$ and optimize the probability parameters $\theta_\sigma$ instead of $w_\sigma$.

Following (4), we replace $\sigma$ by $w_\sigma$ to emphasize the parameters on activation function; we replace $w$ by $\overline{w_\sigma}$ to denote the rest parameters of GNN model. Task objective $\mathcal{M}$ is jointly integrated into stochastic relaxation with space regularization $\mathcal{R}$. The ultimate problem is formulated as below:

$$\theta_\sigma^* = \arg\min_{\theta_\sigma} \left\{ \mathcal{J}(\theta_\sigma) \equiv \mathbb{E}_{w_\sigma \sim p_{\theta_\sigma}(w_\sigma)}[\mathcal{M}(w_\sigma, \overline{w_\sigma}^*; \mathcal{D}_{\text{val}}) + \eta R(w_\sigma)] \right\},$$
$$\text{s.t.} \quad \overline{w_\sigma}^* = \arg\min_{\overline{w_\sigma}} \mathcal{L}(\overline{w_\sigma}, w_\sigma; \mathcal{D}_{\text{train}}),$$

(4)

where $w_\sigma$ denotes the parameters of activation functions $\sigma$ with a regularization term $R$ weighted by $\eta$, $\overline{w_\sigma}$ represents GNN parameters, $\mathcal{L}$ is downstream task criterion of interest, $\mathcal{M}$ is upstream task criterion of interest, probably non-differentiable, $\theta_\sigma$ represents the re-parameterization of $w_\sigma$ through Gaussian distribution, then the whole learning problem is optimized in a stochastic way.

To compute the target loss gradient with respect to probability parameters $\nabla_{\theta_\sigma}\mathcal{J}(\theta_\sigma)$, we have the following proposition. The proof is given in Appendix A.

**Proposition 1** *Let $w_\sigma \sim p_{\theta_\sigma}(w_\sigma)$ represent that the weights of activation functions are sampled from $p_{\theta_\sigma}$. We have*

$$\nabla_{\theta_\sigma}\mathcal{J}(\theta_\sigma) = \nabla_{\theta_\sigma}\mathbb{E}_{w_\sigma \sim p_{\theta_\sigma}(w_\sigma)}[\mathcal{M}(w_\sigma, \overline{w_\sigma}^*; \mathcal{D}_{val}) + \eta R(w_\sigma)]$$
$$= \mathbb{E}_{w_\sigma \sim p_{\theta_\sigma}(w_\sigma)}[[\mathcal{M}(w_\sigma, \overline{w_\sigma}^*; \mathcal{D}_{val}) + \eta R(w_\sigma)]\nabla_{\theta_\sigma}\log p_{\theta_\sigma}(w_\sigma)]$$

(5)

With the help of stochastic relaxation, the previously needed derivation of $\mathcal{M}$ is replaced by a multiplication between forward pass of $\mathcal{M}$ and a gradient of probability loss. In practice, this gradient expectation could be further approximated by Monte Carlo samplings, i.e. $\nabla_{\theta_\sigma}\mathcal{J}(\theta_\sigma) \approx \sum_{i=1}^K \nabla_{\theta_\sigma}\log p_{\theta_\sigma}(w_\sigma^i)[\mathcal{M}(w_\sigma^i, \overline{w_\sigma}^*; \mathcal{D}_{\text{val}}) + \eta R(w_\sigma^i)]$, $K$ is the sample number that we use to approximate the gradient. As a result, the differentiability requirement of $\mathcal{M}$ is removed.

## 4.4 SEARCH STRATEGY

According to (4), the learning is divided in two levels. The outer level optimizes probability parameters $\theta_\sigma$ on validation dataset with (non-differentiable) metric $\mathcal{M}$. Every time $K$ number of samples are generated from the probability distribution (such as Gaussian). Each sample is forwarded and calculated according to (Prop. 1), whose average is an approximation of outer level loss gradient. The optimization of outer level parameters $\theta_\sigma$ influences directly the value of activation function weights since the weights are sample from the updated probability every time a forward pass is needed. The inner level optimizes GNN parameters $\overline{w_\sigma}$ on training dataset with metric $\mathcal{L}$. It is similar to a normal training epoch of any network. The outer and inner levels are interplayed to accelerate convergence.

---

**Algorithm 1** TAFS: Task-aware Activation Function Search

---

1: Initialize $\boldsymbol{\theta}^0 = \mathbf{1}$, initialize $\overline{w}_\sigma$ by Xavier initialization and $w_\sigma$ randomly sampled from $p_{\boldsymbol{\theta}^0}(w_\sigma)$.
2: **for** $m = 0, \ldots, M - 1$ **do**
3:     *// Outer level optimization*
4:     Freeze GNN paramaters $\overline{w}_\sigma$;
5:     **for** $k = 0, \ldots, K - 1$ **do**
6:         Sample activation functions weights $w_\sigma^k$ from $p_{\boldsymbol{\theta}^m}(w_\sigma)$;
7:         Forward inference of the whole network and accumulate stochastic loss $\mathcal{J}(\theta_\sigma)$ as in Prop 1;
8:     **end for**
9:     Obtain $\nabla_{\theta_\sigma} \mathcal{J}(\theta_\sigma)$ by automatic differentiation and update $\theta^m$;
10:    *// Inner level optimization*
11:    Sample activation function paramaters $w_\sigma$ from distribution $p_{\boldsymbol{\theta}^m}(w_\sigma)$ and freeze $w_\sigma$;
12:    Forward inference of the whole network to obtain loss $\mathcal{L}$;
13:    Update GNN parameters $\overline{w}_\sigma$ by automatic differentiation;
14: **end for**
15: Training until convergence and obtain the final model parameter $\overline{w}_\sigma^*$ and dist. parameter $\theta_\sigma$;
16: **return** Final model parameter $\overline{w}_\sigma^*$ and distribution parameter $\theta_\sigma$.

---

**Table 1:** Our proposed TAFS (Task-aware Activation Function Search) enables efficient differentiable search through a flexible and powerful MLP functional space. TAFS supports non-differentiable objective metrics in diverse GNN applications.

| Search Method | Search Efficiency | | Non-Differentiable Metric |
|---|---|---|---|
| | Search Space | Search Strategy | |
| Swish | Discrete template choice | Reinforcement Learning | Applicable |
| APL | Explicit piecewise linear | Differentiable | Not applicable |
| TAFS (ours) | Continuous implicit MLP | Differentiable | Applicable |

The complete TAFS algorithm is given in Algorithm 1. We also compare in Table 1 our proposed TAFS and literature methods. From the time efficiency perspective, Swish is the slowest in Table 1 because it optimizes a new network until convergence before the learning of RL controller. APL on the other hand, has a number of parameters dependent of base models due to its adaptability per neuron. As a result, TAFS enjoys a compact search space without over-parameterization and has superior efficiency in searching. Empirical results are given in Table 4.

## 5 EXPERIMENTS

In this section, we experiment our methods on diverse GNN applications including node classification and link prediction, in order to fully evaluation the methods on differentiable and non-differentiable metrics. Later, we provide detailed analysis on search efficiency and hyperparameter impact. All our experiments are run on single NVIDIA RTX 3090.

### 5.1 NODE CLASSIFICATION

**Datasets.** We experiment on diverse graph datasets for node tasks, including Cora and DBLP for paper classification based on reference network, Cornell and Texas for webpage classification from university network, and Chameleon for wikipedia page classification based on hyperlink network. Statistics are in Appx. E. The task metric here is classification accuracy.

**Baselines.** To fairly compare different activation functions, we compare our searchable activation functions with manually designed ones or previously searched function. Some of these activation functions are visualized in Figure 3(a). For each dataset and baseline chosen, we evaluate on two aggregation layers (GCN and GraphSage) and five network connection topologies (stack, residual, dense, jump knowledge, mixhop). Each model has four layers of aggregation layers. The model is trained for 400 epochs.

**Table 2:** Overall node classification improvement of different models on different datasets. Metric is classification accuracy. Avg. Imp. is the improvement of TAFS with respect to the other choices averaged over all the datasets.

| | Model | Activation | Cora | DBLP | Cornell | Texas | Chameleon | Avg. Imp. |
|---|---|---|---|---|---|---|---|---|
| **GCN** | Stack | ReLU | $83.06_{\pm0.66}$ | $84.63_{\pm0.21}$ | $\mathbf{56.76_{\pm5.92}}$ | $60.54_{\pm6.42}$ | $61.60_{\pm1.75}$ | ↑ 2.8% |
| | | Tanh | $84.82_{\pm0.51}$ | $85.58_{\pm0.15}$ | $\mathbf{56.49_{\pm5.19}}$ | $57.84_{\pm5.01}$ | $61.51_{\pm1.88}$ | ↑ 3.2% |
| | | L-ReLU | $84.57_{\pm0.93}$ | $84.50_{\pm0.40}$ | $\mathbf{57.38_{\pm2.16}}$ | $60.54_{\pm7.37}$ | $61.95_{\pm2.18}$ | ↑ 2.1% |
| | | Swish | $83.88_{\pm0.81}$ | $84.89_{\pm0.34}$ | $57.30_{\pm3.97}$ | $58.65_{\pm5.55}$ | $58.33_{\pm1.68}$ | ↑ 4.1% |
| | | **TAFS** | $\mathbf{89.08_{\pm0.48}}$ | $\mathbf{86.24_{\pm0.17}}$ | $57.37_{\pm4.37}$ | $\mathbf{62.11_{\pm5.48}}$ | $\mathbf{62.31_{\pm1.82}}$ | - |
| | Residual | ReLU | $85.13_{\pm0.95}$ | $84.45_{\pm0.34}$ | $57.84_{\pm5.43}$ | $57.84_{\pm5.95}$ | $66.93_{\pm2.17}$ | ↑ 3.2% |
| | | Tanh | $86.02_{\pm0.55}$ | $85.63_{\pm0.14}$ | $\mathbf{58.38_{\pm4.39}}$ | $57.57_{\pm5.93}$ | $68.86_{\pm1.84}$ | ↑ 2.0% |
| | | L-ReLU | $86.60_{\pm0.72}$ | $84.97_{\pm0.33}$ | $55.68_{\pm8.30}$ | $57.84_{\pm6.75}$ | $67.50_{\pm1.48}$ | ↑ 3.3% |
| | | Swish | $85.86_{\pm0.64}$ | $84.67_{\pm0.19}$ | $56.22_{\pm6.14}$ | $\mathbf{60.54_{\pm7.66}}$ | $66.29_{\pm2.12}$ | ↑ 2.8% |
| | | **TAFS** | $\mathbf{88.16_{\pm0.58}}$ | $\mathbf{86.29_{\pm0.18}}$ | $58.20_{\pm4.80}$ | $60.22_{\pm5.51}$ | $\mathbf{70.49_{\pm1.64}}$ | - |
| | JKNet | ReLU | $86.86_{\pm0.71}$ | $84.99_{\pm0.25}$ | $76.49_{\pm7.36}$ | $77.57_{\pm7.36}$ | $58.18_{\pm1.63}$ | ↑ 3.8% |
| | | Tanh | $86.41_{\pm0.57}$ | $85.57_{\pm0.20}$ | $68.92_{\pm6.76}$ | $65.95_{\pm9.22}$ | $\mathbf{60.20_{\pm2.19}}$ | ↑ 9.1% |
| | | L-ReLU | $87.45_{\pm0.51}$ | $85.04_{\pm0.15}$ | $74.05_{\pm5.57}$ | $76.49_{\pm8.80}$ | $57.98_{\pm2.36}$ | ↑ 4.1% |
| | | Swish | $86.34_{\pm0.92}$ | $84.95_{\pm0.28}$ | $77.03_{\pm5.57}$ | $78.11_{\pm6.99}$ | $57.00_{\pm2.54}$ | ↑ 4.7% |
| | | **TAFS** | $\mathbf{88.84_{\pm0.56}}$ | $\mathbf{87.07_{\pm0.22}}$ | $\mathbf{81.35_{\pm6.40}}$ | $\mathbf{81.08_{\pm5.01}}$ | $60.21_{\pm2.04}$ | - |
| | Mixhop | ReLU | $85.31_{\pm0.64}$ | $85.10_{\pm0.18}$ | $73.78_{\pm5.55}$ | $74.05_{\pm9.53}$ | $51.64_{\pm2.24}$ | ↑ 2.7% |
| | | Tanh | $85.15_{\pm0.67}$ | $85.12_{\pm0.30}$ | $72.97_{\pm7.55}$ | $76.76_{\pm6.86}$ | $50.59_{\pm2.60}$ | ↑ 2.6% |
| | | L-ReLU | $86.38_{\pm0.50}$ | $85.01_{\pm0.17}$ | $72.43_{\pm6.14}$ | $72.70_{\pm5.05}$ | $51.36_{\pm2.80}$ | ↑ 3.4% |
| | | Swish | $86.21_{\pm1.03}$ | $85.43_{\pm0.25}$ | $72.34_{\pm8.18}$ | $74.86_{\pm6.51}$ | $51.89_{\pm2.10}$ | ↑ 2.5% |
| | | **TAFS** | $\mathbf{88.77_{\pm0.57}}$ | $\mathbf{86.18_{\pm0.17}}$ | $\mathbf{75.14_{\pm5.38}}$ | $\mathbf{78.43_{\pm5.28}}$ | $\mathbf{52.17_{\pm1.97}}$ | - |
| **GraphSage** | Stack | ReLU | $83.06_{\pm0.66}$ | $83.67_{\pm0.41}$ | $58.11_{\pm6.19}$ | $70.00_{\pm6.78}$ | $47.02_{\pm4.20}$ | ↑ 12.1% |
| | | Tanh | $84.82_{\pm0.51}$ | $84.90_{\pm0.19}$ | $68.65_{\pm6.75}$ | $71.89_{\pm7.85}$ | $53.50_{\pm1.68}$ | ↑ 4.3% |
| | | L-ReLU | $84.57_{\pm0.65}$ | $84.16_{\pm0.23}$ | $62.16_{\pm5.92}$ | $68.11_{\pm7.23}$ | $49.21_{\pm3.02}$ | ↑ 9.8% |
| | | Swish | $81.53_{\pm0.74}$ | $83.62_{\pm0.50}$ | $57.03_{\pm6.45}$ | $68.65_{\pm6.19}$ | $48.42_{\pm2.17}$ | ↑ 13.0% |
| | | **TAFS** | $\mathbf{87.08_{\pm0.48}}$ | $\mathbf{85.22_{\pm0.30}}$ | $\mathbf{72.43_{\pm7.23}}$ | $\mathbf{74.51_{\pm6.92}}$ | $\mathbf{58.57_{\pm1.20}}$ | - |
| | Residual | ReLU | $84.11_{\pm0.82}$ | $83.05_{\pm0.33}$ | $65.95_{\pm6.64}$ | $73.51_{\pm6.71}$ | $55.02_{\pm2.73}$ | ↑ 6.1% |
| | | Tanh | $85.62_{\pm0.52}$ | $85.22_{\pm0.17}$ | $72.43_{\pm3.97}$ | $\mathbf{78.11_{\pm9.00}}$ | $59.17_{\pm1.80}$ | ↑ 0.6% |
| | | L-ReLU | $85.63_{\pm0.42}$ | $84.05_{\pm0.21}$ | $71.89_{\pm3.67}$ | $74.86_{\pm5.80}$ | $55.86_{\pm1.83}$ | ↑ 3.1% |
| | | Swish | $84.97_{\pm0.79}$ | $84.17_{\pm0.43}$ | $71.08_{\pm4.02}$ | $75.41_{\pm7.09}$ | $54.17_{\pm1.44}$ | ↑ 3.9% |
| | | **TAFS** | $\mathbf{89.10_{\pm0.385}}$ | $\mathbf{85.22_{\pm0.17}}$ | $\mathbf{73.38_{\pm3.63}}$ | $77.03_{\pm6.86}$ | $58.62_{\pm2.08}$ | - |
| | JKNet | ReLU | $85.29_{\pm0.56}$ | $83.97_{\pm0.15}$ | $80.00_{\pm6.07}$ | $81.62_{\pm5.10}$ | $56.78_{\pm1.62}$ | ↑ 3.0% |
| | | Tanh | $86.01_{\pm0.51}$ | $85.25_{\pm0.18}$ | $77.03_{\pm5.57}$ | $78.92_{\pm6.14}$ | $57.68_{\pm1.92}$ | ↑ 3.8% |
| | | L-ReLU | $85.90_{\pm0.42}$ | $85.01_{\pm0.25}$ | $80.27_{\pm6.84}$ | $81.35_{\pm4.75}$ | $57.41_{\pm2.01}$ | ↑ 2.5% |
| | | Swish | $85.56_{\pm0.61}$ | $84.71_{\pm0.22}$ | $77.13_{\pm5.30}$ | $81.06_{\pm5.43}$ | $55.00_{\pm1.93}$ | ↑ 4.5% |
| | | **TAFS** | $\mathbf{89.51_{\pm0.66}}$ | $\mathbf{86.73_{\pm0.20}}$ | $\mathbf{81.79_{\pm5.08}}$ | $\mathbf{82.10_{\pm5.16}}$ | $\mathbf{59.37_{\pm1.53}}$ | - |
| | Mixhop | ReLU | $84.55_{\pm1.07}$ | $84.23_{\pm0.21}$ | $75.95_{\pm9.59}$ | $81.63_{\pm4.80}$ | $54.19_{\pm2.15}$ | ↑ 2.3% |
| | | Tanh | $84.82_{\pm0.65}$ | $84.21_{\pm0.38}$ | $\mathbf{78.11_{\pm6.99}}$ | $81.89_{\pm5.80}$ | $53.20_{\pm1.21}$ | ↑ 2.0% |
| | | L-ReLU | $84.80_{\pm1.10}$ | $84.34_{\pm0.28}$ | $76.22_{\pm8.36}$ | $77.30_{\pm4.71}$ | $53.14_{\pm1.72}$ | ↑ 3.7% |
| | | Swish | $84.18_{\pm0.55}$ | $84.69_{\pm0.30}$ | $75.95_{\pm8.15}$ | $80.27_{\pm6.74}$ | $53.20_{\pm1.86}$ | ↑ 3.0% |
| | | **TAFS** | $\mathbf{87.77_{\pm1.40}}$ | $\mathbf{85.30_{\pm0.24}}$ | $77.77_{\pm4.39}$ | $\mathbf{83.70_{\pm4.05}}$ | $\mathbf{55.07_{\pm0.57}}$ | - |

**Results.** We provide in Table 2 the results of node classification tasks. The improvement of TAFS with respect to the other function choices is significant. Note that the improvements are observable across different graph data and GNN models, showing that TAFS is task-aware to graphs in citation, university webpage, wikipedia link graph, etc.

## 5.2 MOLECULE AND PROTEIN INTERACTION PREDICTION

**Datasets.** Biomedical graph are one of the most active and effective application areas of GNN. Biomedical GNN has accelerated important studies in protein prediction, molecule generation, gene

expression, etc. We consider the link prediction that is a typical task in molecule and protein interaction prediction. Specifically, we consider Drug-Drug Interaction (DDI), Drug-Target Interaction (DTI), Protein-Protein Interaction (PPI) and Disea-Gene Association (DGA). The statistics of the four datasets are provided in Appx. Table 6.

**Baselines.** We adopt two biomedical graph baselines SkipGNN (Huang et al., 2020) and HOGCN (KC et al., 2022). SkipGNN proposes a general GNN architecture to model molecular interactions and works well on all these biomedical tasks. We use both as base model and experiment TAFS to replace the activation functions. The training hyperparameters are the same as in the orginal work.

**Table 3:** Drug and protein interaction predictions.

| Task | Model | Activation | ROCAUC | PRAUC |
|------|-------|-----------|--------|-------|
| Drug-Target Interaction | SkipGNN | ReLU | $0.922_{\pm0.004}$ | $0.928_{\pm0.006}$ |
| | | TAFS w.o. relaxation | $0.933_{\pm0.002}$ | $0.934_{\pm0.001}$ |
| | | **TAFS** | $\mathbf{0.952_{\pm0.001}}$ | $\mathbf{0.954_{\pm0.001}}$ |
| | HOGCN | ReLU | $0.927_{\pm0.001}$ | $0.929_{\pm0.001}$ |
| | | TAFS w.o. relaxation | $0.923_{\pm0.002}$ | $0.922_{\pm0.001}$ |
| | | **TAFS** | $\mathbf{0.943_{\pm0.002}}$ | $\mathbf{0.940_{\pm0.001}}$ |
| Drug-Drug Interaction | SkipGNN | ReLU | $0.886_{\pm0.003}$ | $0.866_{\pm0.006}$ |
| | | TAFS w.o. relaxation | $0.890_{\pm0.002}$ | $0.874_{\pm0.001}$ |
| | | **TAFS** | $\mathbf{0.911_{\pm0.002}}$ | $\mathbf{0.898_{\pm0.003}}$ |
| | HOGCN | ReLU | $0.898_{\pm0.002}$ | $0.881_{\pm0.003}$ |
| | | TAFS w.o. relaxation | $0.897_{\pm0.002}$ | $0.901_{\pm0.002}$ |
| | | **TAFS** | $\mathbf{0.917_{\pm0.002}}$ | $\mathbf{0.901_{\pm0.001}}$ |
| Protein-Protein Interaction | SkipGNN | ReLU | $0.917_{\pm0.004}$ | $0.921_{\pm0.003}$ |
| | | TAFS w.o. relaxation | $0.920_{\pm0.001}$ | $0.922_{\pm0.002}$ |
| | | **TAFS** | $\mathbf{0.927_{\pm0.001}}$ | $\mathbf{0.937_{\pm0.002}}$ |
| | HOGCN | ReLU | $0.919_{\pm0.001}$ | $0.922_{\pm0.002}$ |
| | | TAFS w.o. relaxation | $0.919_{\pm0.002}$ | $0.924_{\pm0.001}$ |
| | | **TAFS** | $\mathbf{0.923_{\pm0.003}}$ | $\mathbf{0.929_{\pm0.002}}$ |
| Disease-Gene Association | SkipGNN | ReLU | $0.912_{\pm0.004}$ | $0.915_{\pm0.003}$ |
| | | TAFS w.o. relaxation | $0.916_{\pm0.001}$ | $0.920_{\pm0.001}$ |
| | | **TAFS** | $\mathbf{0.930_{\pm0.001}}$ | $\mathbf{0.940_{\pm0.001}}$ |
| | HOGCN | ReLU | $0.927_{\pm0.001}$ | $0.934_{\pm0.001}$ |
| | | TAFS w.o. relaxation | $0.929_{\pm0.002}$ | $0.933_{\pm0.001}$ |
| | | **TAFS** | $\mathbf{0.933_{\pm0.001}}$ | $\mathbf{0.942_{\pm0.002}}$ |

**Results.** We provide in Table 3 the results of four link prediction tasks. Again, both SkipGNN and HOGCN use ReLU by default. With TAFS, SkipGNN and HOGCN has gained significant performance evaluated in ROCAUC and PRAUC, two non-differentiable metrics. Furthermore, when TAFS is integrated with SkipGNN, a model from 2020, it outperforms HOGCN, the state-of-the-art model from 2022. This underscores the significance of activation function search, which has hitherto been overlooked in the GNN community.

### 5.3 ABLATION STUDY

To further analyze our proposed TAFS algorithm, we provide additional experiments to illustrate the search results, search efficiency and hyperparameter impact.

**Visualization of activation function search.** We show in Figure 3 the searched activation functions from the literature methods and TAFS. It can be observed that TAFS could find diverse activation functions different than manually design ones or the searched ones by Swish and APL. Moreover, TAFS learns layer-wise activation function, leading to different behaviours of functions in different layers as in Figure 3(b)(c). Deeper layers' activation functions are smoother than shallow layers.

**Search efficiency** The modeling differences between literature search methods and TAFS are given in Table 1. In this part, we provide more empirical details of the search efficiency comparison in Table 4. TAFS has a significantly smaller consumption of extra memory and shorter running time. This huge efficiency improvement is credited to TAFS' compact MLP functional search space and differentiable search strategy, making TAFS' extra parameters *independent* of base models, whichever dataset or

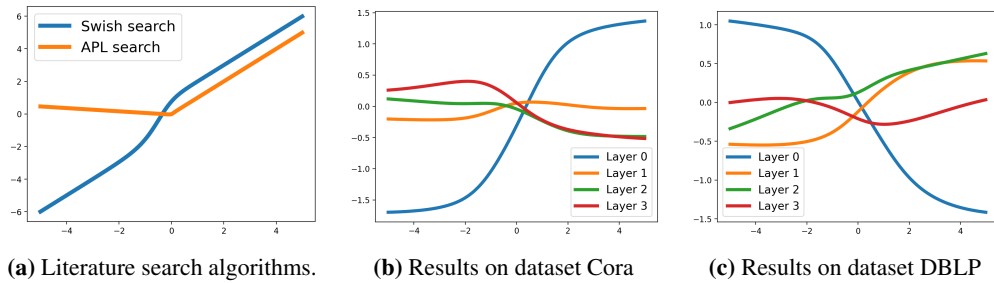

**(a)** Literature search algorithms.  **(b)** Results on dataset Cora  **(c)** Results on dataset DBLP

**Figure 3:** Literature activation functions and searched activation results. (b)(c) include searched results on different layers across two datasets.

GNN model (as long as the model has the same number of activation functions), while APL models each neuron with a piecewise linear unit, leading to over *2000 times* more parameters than TAFS.

**Hyperparameters impact.** The choice of hyperparameters significantly affects performance. TAFS introduces two sets of hyperparameters: the number of samples (K) and the selection of the MLP architecture. We present their effects in Figure 4. The number of samples (K) in stochastic optimization exhibits a consistent increasing trend, representing a trade-off between accuracy and computational time. Regarding MLP hyperparameters, we analyze their impact on two node tasks, DBLP and Cornell, using nine different configurations: depths ranging from two to four layers and widths spanning from 10 to 1000 neurons. It is evident that a very small MLP (e.g., two layers with 10 neurons) is inadequate for

**Table 4:** Search efficiency comparison.

| Dataset | Model | Parameters | Time(min) |
|---|---|---|---|
| DBLP | Base | 420K | 0.15 |
| | Swish | +340K (+82%) | 350 |
| | APL | +2400K (+575%) | 4 |
| | **TAFS** | **+1.3K** (+0.3%) | **1** |
| Chameleon | Base | 315K | 1.2 |
| | Swish | +420K (+108%) | 2990 |
| | APL | +1760K (+558%) | 33 |
| | **TAFS** | **+1.3K** (+0.4%) | **11** |
| Ogbg-Molhiv | Base | 27M | 1020 |
| | Swish | - | > 70 days |
| | APL | OOM (+150M) | - |
| | **TAFS** | **+12K** | **1380** |

modeling adaptive activation functions. However, the distinctions between other choices are negligible. Given that deeper and wider MLPs require significantly more parameters, we opt for a two-layer MLP with 100 hidden units in all other experiments.

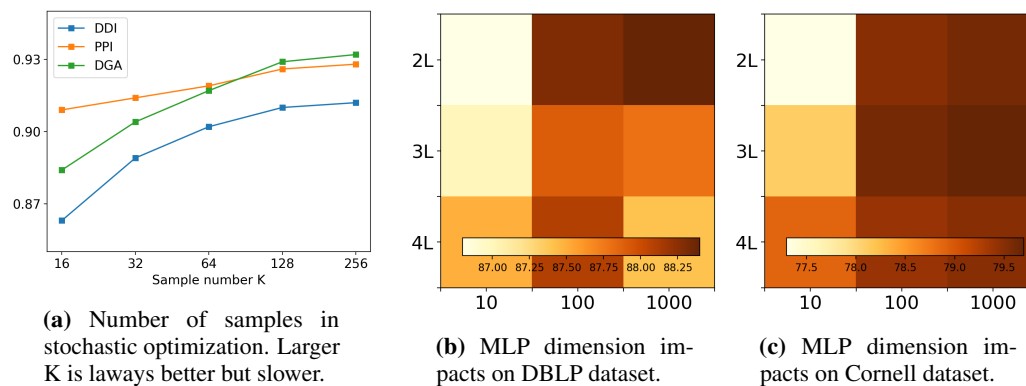

**(a)** Number of samples in stochastic optimization. Larger K is laways better but slower.

**(b)** MLP dimension impacts on DBLP dataset.

**(c)** MLP dimension impacts on Cornell dataset.

**Figure 4:** Hyperparameter impact of number of samples in stochastic relaxation and the impact of MLP dimensions. In (b)(c), deeper color means better performance.

## 6 CONCLUSION

In a word, we achieve a task-aware activation function search in GNN through an expressive and compact representation of search space, stochastic relaxation with reparameterization, which are carefully co-designed with search strategy. Our search space is inclusive and parameter efficient, including appropriate number of high-quality functions. The search strategy is end-to-end trained and every operation of the framework is differentiable. Finally, the stochastic relaxation is capable of dealing with any metric of interest, closing the optimization gap.

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

## A  Proof of Proposition 1

$$
\begin{aligned}
\nabla_{\theta_\sigma} \mathcal{J}(\theta_\sigma) &= \nabla_{\theta_\sigma} \mathbb{E}_{w_\sigma \sim p_{\theta_\sigma}(w_\sigma)}[\mathcal{M}(w_\sigma, \overline{w_\sigma}^*; \mathcal{D}_{\text{val}}) + \eta R(w_\sigma)] \\
&= \int \nabla_{\theta_\sigma} p_{\theta_\sigma}(w_\sigma)[\mathcal{M}(w_\sigma, \overline{w_\sigma}^*; \mathcal{D}_{\text{val}}) + \eta R(w_\sigma)] dw_\sigma \\
&= \int [\mathcal{M}(w_\sigma, \overline{w_\sigma}^*; \mathcal{D}_{\text{val}}) + \eta R(w_\sigma)] \frac{\nabla_{\theta_\sigma} p_{\theta_\sigma}(w_\sigma)}{p_{\theta_\sigma}(w_\sigma)} p_{\theta_\sigma}(w_\sigma) dw_\sigma \\
&= \int [\mathcal{M}(w_\sigma, \overline{w_\sigma}^*; \mathcal{D}_{\text{val}}) + \eta R(w_\sigma)] p_{\theta_\sigma}(w_\sigma) \nabla_{\theta_\sigma} \log p_{\theta_\sigma}(w_\sigma) dw_\sigma \\
&= \mathbb{E}_{w_\sigma \sim p_{\theta_\sigma}(w_\sigma)}[[\mathcal{M}(w_\sigma, \overline{w_\sigma}^*; \mathcal{D}_{\text{val}}) + \eta R(w_\sigma)] \nabla_{\theta_\sigma} \log p_{\theta_\sigma}(w_\sigma)] \\
&\approx \sum_{i=1}^{K} \nabla_{\theta_\sigma} \log p_{\theta_\sigma}(w_\sigma^i)[\mathcal{M}(w_\sigma^i, \overline{w_\sigma}^*; \mathcal{D}_{\text{val}}) + \eta R(w_\sigma^i)]
\end{aligned}
\tag{6}
$$

## B  More details about Swish and APL

Two notable works have set up to solve this problem. Adaptive piecewise linear (APL) (Agostinelli et al., 2015) parameterizes $\sigma$ of each neuron by a sum of hinge-shaped functions $\sigma(x) = \max(0, x) + \Sigma_{s=1}^{S} a^s \max(0, -x + b^s)$, leading to a piecewise linear function with $S$ a predefined hyperparameter. $a$ and $b$ are learnable parameters to control the slope and location of each hinge. Swish (Ramachandran et al., 2018) takes the idea of Neural Architecture Search (NAS) and proposes to search through a symbolic discrete space composing of unary and binary functions. Typical mathematical operations are included such as (unary) $x, -x, \beta x, \sin x, \cos x, \tanh x, \tanh^{-1} x, \exp x, \log(1 + \exp(x))$ and (binary) $x_1 + x_2, x_1 - x_2, x_1 \times x_2, \max(x_1, x_2), \min(x_1, x_2), \sigma(x_1)x_2$, etc.

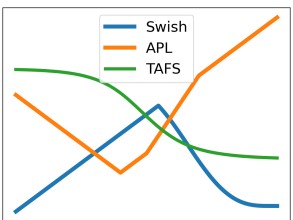

**Figure 5:** Search space illustration.

APL's piecewise linear function space, in theory, can approximate any functions of interest with appropriate choice of hinge numbers. However, this is in practice unrealizable because we do not know how much complexity the activation pattern requires. For example, in the authors' experiment, they choose $S = 2$ in the case of Cifar-100, which means basically all the functions learned are just two straight lines, similar to ReLU. There are many functions *out of the scope* of this space, such as Tanh, which we show significantly outperform the others in Figure 1. Moreover, APL learns activation function per hidden unit, which introduces too many parameters. Total number of extra parameters in APL is 2SN, where S is number of hinges and N is number of hidden units.

Swish has a larger space since it contains atomic trigonometric functions. But still, all the searched functions are explicitly written as cascading the atomic operations while many other functions cannot satisfy this requirement. Moreover, Swish uses policy gradient to train a controller that optimize the decision choices in the space, which is very *inefficient* compared to end-to-end differentiable search. Lastly, APL fails to optimize towards non-differentiable metrics, which are quite common in especially graph data such as ROCAUC for link prediction. On the other hand, Swish do not have such limitation since the reinforcement learning algorithm is intrinsically adaptive to all metrics.

Besides Swish and APL, our work has also been inspired by Network in Network (NiN) (Lin et al., 2013). Classic convolutional operation applies linear filters to extract local features. NiN replaces a linear filter by non-linear MLP for better expressiveness, i.e. MLPConv. We note that it has as inputs all the elements within the receptive field, making it non-univariate. Thus we adopt similar principle, i.e., using generalizable approximators to replace low complexity modules in a network.

## C MORE BACKGROUND ON LIPSCHITZ SMOOTHNESS

Lipschitz smoothness has been explored in many deep learning works. Specifically, we have the following formal definition and proposition.

**Definition 1** *(Lipschitz Smoothness(Weng et al., 2018)) Let $S \in \mathrm{R}^d$ be a convex bounded closed set and let $h(x) : S \to R$ be a continuously differentiable function on an open set containing $S$. Then $h(x)$ is a Lipschitz funciton with Lipschitz constant $c$ if the following inequality holds for any $x, y \in S$:*

$$|h(x) - h(y)| \leq c||x - y||_p \tag{7}$$

*where $c = \max\{||\nabla h(x)||_q : x \in S\}$, $||\nabla h(x)||_q$ is the gradient norm of $h(x)$, and $1 \leq q \leq \infty$.*

It has been shown that determining the Lipschitz constant of an MLP is NP-hard. As a result, we could not explicitly calculate the Lipschitz constant $c$ and use it as a differentiable metric for optimization. Instead, without loss of generality, we apply Jacobian regularization to improve the smoothness of activation functions.

**Proposition 2** *(Jacobian regularization(Hoffman et al., 2019)) Denote Jacobian matrix $J_{i,j}(x) = \frac{\partial h_i}{\partial x_j}(x)$. Jacobian regularization can be realized by minimizing the additional term $||J(x)||_F = \{\Sigma_{i,j}[J_{i,j}(x)]^2\}$*

The Jacobian minimization term could be combined with any downsteam task such that Eq. 4 applies to activation function search in general learning tasks.

## D CONNECTIONS TO GNN NEURAL ARCHITECTURE SEARCH

Our work focuses on searching the most suitable activation functions without human intervention. This principle is similar to Neural Architecture Search (NAS), which searches the GNN architecture including graph convolution, residual connection, pooling, etc. A subsequent question is why not search activation function together with GNN architectures? The reasons are two-fold. Firstly, we want to emphasize the importance of activation functions in GNN which is largely neglected in current GNN research. It is more appropriate to study this single part in a disentangled way. Secondly, if combined with whole architecture search, the whole search space is significantly larger and it is hard to reduce the space size. In our work, since we have some prior knowledge about the activation functions such as smoothness, we could regularize greatly the space. However, such priors are not easily transferable to other searchable parts.

## E STATISTICS OF DIVERSE EXPERIMENTAL DATASETS

We provide in Table 5 and Table 6 the statistics of dataset used in node and link tasks.

**Table 5:** Statistics of the node datasets

| Datasets | #Nodes | #Edges | #Features | #Classes |
|----------|--------|--------|-----------|----------|
| Cora | 2708 | 5278 | 1433 | 7 |
| DBLP | 17,716 | 105,734 | 1,639 | 4 |
| Cornell | 183 | 280 | 1703 | 5 |
| Texas | 183 | 295 | 1703 | 5 |
| Chameleon | 2277 | 31421 | 2325 | 5 |

**Table 6:** Biomedical tasks and dataset statistics.

| Task | Data Source | #Nodes | #Edges |
|------|-------------|--------|--------|
| Drug-Target Interaction | BIOSNAP | 7343 | 15139 |
| Drug-Drug Interaction | BIOSNAP | 1514 | 48514 |
| Protein-Protein Interaction | HuRI | 5604 | 23322 |
| Disease-Gene Association | DisGeNET | 19783 | 81746 |

## F  MORE EXPERIMENTAL RESULTS

In this section, we include in Table 7 more results on node- and link-level tasks that are removed from the main text due to space limit.

## G  MORE VISUALIZATIONS OF ACTIVATION FUNCTIONS

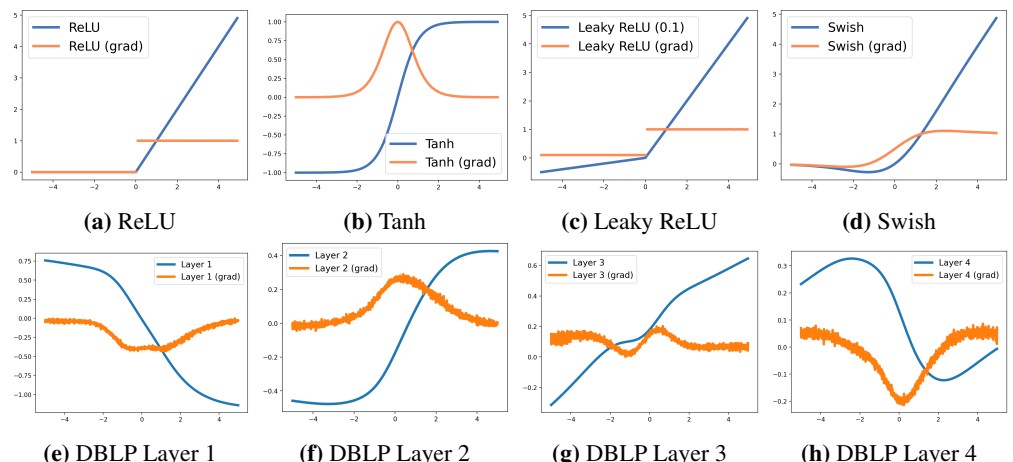

**(a)** ReLU  **(b)** Tanh  **(c)** Leaky ReLU  **(d)** Swish

**(e)** DBLP Layer 1  **(f)** DBLP Layer 2  **(g)** DBLP Layer 3  **(h)** DBLP Layer 4

**Figure 6:** Literature activation functions and searched activation results. (a)-(d) are manually designed activation functions. (e)-(h) are TAFS searched activation functions.

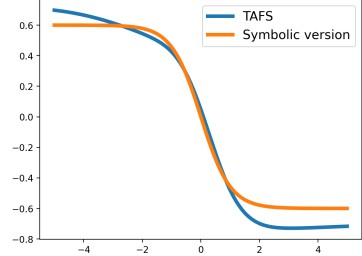

**Figure 7:** Visualization of symbolized formula

| Dataset | Baseline | Activation | Accuracy |
|---------|----------|-----------|----------|
| Cora | GCN-JK | TAFS | 89.08 |
| | | 0.6*Tanh(-x) | 87.89 |

**Table 8:** Comparison with symbolized formula

## H  SYMBOLIZED EXPLICIT FORMULA

On the Cora dataset with GCN-JK network baseline, the searched result is the blue line. We distill an explicit formula by symbolic regression: y = 0.6 Tanh(-x) and plug in this activation function back to the model. The performance is shown in Table 8 and we visualize the function in Figure 7. There is a performance gap in the table below. This is because explicit symbolic space is not accurate, which shows further that our implicit functional space is expressive.

## I   LICENSE OF ASSETS

The source code will be shared under MIT license. All the datasets used in this research is publicly available.

**Table 7:** Overall node classification improvement of different models on different datasets.

| Model | Activation | Cora | DBLP | Cornell | Texas | Chameleon |
|---|---|---|---|---|---|---|
| GCN-Stack | ReLU | $83.06_{\pm0.66}$ | $84.63_{\pm0.21}$ | $56.76_{\pm5.92}$ | $60.54_{\pm6.42}$ | $61.60_{\pm1.75}$ |
| | Tanh | $84.82_{\pm0.51}$ | $85.58_{\pm0.15}$ | $56.49_{\pm5.19}$ | $57.84_{\pm5.01}$ | $61.51_{\pm1.88}$ |
| | LeakyReLU | $84.57_{\pm0.93}$ | $84.50_{\pm0.40}$ | $\mathbf{57.38_{\pm2.16}}$ | $60.54_{\pm7.37}$ | $61.95_{\pm2.18}$ |
| | Swish | $83.88_{\pm0.81}$ | $84.89_{\pm0.34}$ | $57.30_{\pm3.97}$ | $58.65_{\pm5.55}$ | $58.33_{\pm1.68}$ |
| | **TAFS** | $\mathbf{89.08_{\pm0.48}}$ | $\mathbf{86.24_{\pm0.17}}$ | $57.37_{\pm4.37}$ | $\mathbf{62.11_{\pm5.48}}$ | $\mathbf{62.31_{\pm1.82}}$ |
| GCN-Residual | ReLU | $85.13_{\pm0.95}$ | $84.45_{\pm0.34}$ | $57.84_{\pm5.43}$ | $57.84_{\pm5.95}$ | $66.93_{\pm2.17}$ |
| | Tanh | $86.02_{\pm0.55}$ | $85.63_{\pm0.14}$ | $\mathbf{58.38_{\pm4.39}}$ | $57.57_{\pm5.93}$ | $68.86_{\pm1.84}$ |
| | LeakyReLU | $86.60_{\pm0.72}$ | $84.97_{\pm0.33}$ | $55.68_{\pm8.30}$ | $57.84_{\pm6.75}$ | $67.50_{\pm1.48}$ |
| | Swish | $85.86_{\pm0.64}$ | $84.67_{\pm0.19}$ | $56.22_{\pm6.14}$ | $\mathbf{60.54_{\pm7.66}}$ | $66.29_{\pm2.12}$ |
| | **TAFS** | $\mathbf{88.16_{\pm0.58}}$ | $\mathbf{86.29_{\pm0.18}}$ | $58.20_{\pm4.80}$ | $60.22_{\pm5.51}$ | $\mathbf{70.49_{\pm1.64}}$ |
| GCN-Dense | ReLU | $85.88_{\pm0.41}$ | $84.59_{\pm0.38}$ | $53.78_{\pm9.16}$ | $57.30_{\pm6.37}$ | $65.81_{\pm2.12}$ |
| | Tanh | $86.23_{\pm0.55}$ | $85.62_{\pm0.25}$ | $58.65_{\pm5.41}$ | $57.57_{\pm4.69}$ | $67.24_{\pm2.11}$ |
| | LeakyReLU | $86.60_{\pm0.36}$ | $84.68_{\pm0.23}$ | $57.57_{\pm4.37}$ | $58.11_{\pm7.38}$ | $65.66_{\pm1.61}$ |
| | Swish | $86.12_{\pm0.46}$ | $84.77_{\pm0.28}$ | $57.52_{\pm5.68}$ | $59.46_{\pm6.22}$ | $65.94_{\pm2.69}$ |
| | **TAFS** | $\mathbf{89.58_{\pm0.46}}$ | $\mathbf{86.62_{\pm0.16}}$ | $\mathbf{60.27_{\pm4.84}}$ | $\mathbf{61.62_{\pm6.26}}$ | $\mathbf{67.34_{\pm2.24}}$ |
| GCN-JKNet | ReLU | $86.86_{\pm0.71}$ | $84.99_{\pm0.25}$ | $76.49_{\pm7.36}$ | $77.57_{\pm7.36}$ | $58.18_{\pm1.63}$ |
| | Tanh | $86.41_{\pm0.57}$ | $85.57_{\pm0.20}$ | $68.92_{\pm6.76}$ | $65.95_{\pm9.22}$ | $60.20_{\pm2.19}$ |
| | LeakyReLU | $87.45_{\pm0.51}$ | $85.04_{\pm0.15}$ | $74.05_{\pm5.57}$ | $76.49_{\pm8.80}$ | $57.98_{\pm2.36}$ |
| | Swish | $86.34_{\pm0.92}$ | $84.95_{\pm0.28}$ | $77.03_{\pm5.57}$ | $78.11_{\pm6.99}$ | $57.00_{\pm2.54}$ |
| | **TAFS** | $\mathbf{88.84_{\pm0.56}}$ | $\mathbf{87.07_{\pm0.22}}$ | $\mathbf{81.35_{\pm6.40}}$ | $\mathbf{81.08_{\pm5.01}}$ | $\mathbf{60.21_{\pm2.04}}$ |
| GCN-Mixhop | ReLU | $85.31_{\pm0.64}$ | $85.10_{\pm0.18}$ | $73.78_{\pm5.55}$ | $74.05_{\pm9.53}$ | $51.64_{\pm2.24}$ |
| | Tanh | $85.15_{\pm0.67}$ | $85.12_{\pm0.30}$ | $72.97_{\pm7.55}$ | $76.76_{\pm6.86}$ | $50.59_{\pm2.60}$ |
| | LeakyReLU | $86.38_{\pm0.50}$ | $85.01_{\pm0.17}$ | $72.43_{\pm6.14}$ | $72.70_{\pm5.05}$ | $51.36_{\pm2.80}$ |
| | Swish | $86.21_{\pm1.03}$ | $85.43_{\pm0.25}$ | $72.34_{\pm8.18}$ | $74.86_{\pm6.51}$ | $51.89_{\pm2.10}$ |
| | **TAFS** | $\mathbf{88.77_{\pm0.57}}$ | $\mathbf{86.18_{\pm0.17}}$ | $\mathbf{75.14_{\pm5.38}}$ | $\mathbf{78.03_{\pm7.49}}$ | $\mathbf{52.17_{\pm1.97}}$ |
| Sage-Stack | ReLU | $83.06_{\pm0.66}$ | $83.67_{\pm0.41}$ | $58.11_{\pm6.19}$ | $70.00_{\pm6.78}$ | $47.02_{\pm4.20}$ |
| | Tanh | $84.82_{\pm0.51}$ | $84.90_{\pm0.19}$ | $68.65_{\pm6.75}$ | $71.89_{\pm7.85}$ | $53.50_{\pm1.68}$ |
| | LeakyReLU | $84.57_{\pm0.65}$ | $84.16_{\pm0.23}$ | $62.16_{\pm5.92}$ | $68.11_{\pm7.23}$ | $49.21_{\pm3.02}$ |
| | Swish | $81.53_{\pm0.74}$ | $83.62_{\pm0.50}$ | $57.03_{\pm6.45}$ | $68.65_{\pm6.19}$ | $48.42_{\pm2.17}$ |
| | **TAFS** | $\mathbf{87.08_{\pm0.48}}$ | $\mathbf{85.22_{\pm0.30}}$ | $\mathbf{72.43_{\pm7.23}}$ | $\mathbf{74.51_{\pm6.92}}$ | $\mathbf{58.57_{\pm1.20}}$ |
| Sage-Residual | ReLU | $84.11_{\pm0.82}$ | $83.05_{\pm0.33}$ | $65.95_{\pm6.64}$ | $73.51_{\pm6.71}$ | $55.02_{\pm2.73}$ |
| | Tanh | $85.62_{\pm0.52}$ | $85.22_{\pm0.17}$ | $72.43_{\pm3.97}$ | $\mathbf{78.11_{\pm9.00}}$ | $\mathbf{59.17_{\pm1.80}}$ |
| | LeakyReLU | $85.63_{\pm0.42}$ | $84.05_{\pm0.21}$ | $71.89_{\pm3.67}$ | $74.86_{\pm5.80}$ | $55.86_{\pm1.83}$ |
| | Swish | $84.97_{\pm0.79}$ | $84.17_{\pm0.43}$ | $71.08_{\pm4.02}$ | $75.41_{\pm7.09}$ | $54.17_{\pm1.44}$ |
| | **TAFS** | $\mathbf{89.10_{\pm0.385}}$ | $\mathbf{85.22_{\pm0.17}}$ | $\mathbf{73.38_{\pm3.63}}$ | $77.03_{\pm6.86}$ | $58.62_{\pm2.08}$ |
| Sage-Dense | ReLU | $84.65_{\pm0.66}$ | $83.03_{\pm0.39}$ | $77.03_{\pm8.98}$ | $76.49_{\pm6.95}$ | $58.00_{\pm1.29}$ |
| | Tanh | $85.49_{\pm1.04}$ | $85.15_{\pm0.21}$ | $75.14_{\pm5.77}$ | $78.92_{\pm6.14}$ | $\mathbf{58.29_{\pm1.63}}$ |
| | LeakyReLU | $85.64_{\pm0.78}$ | $84.24_{\pm0.16}$ | $76.22_{\pm6.14}$ | $74.32_{\pm6.97}$ | $57.17_{\pm2.05}$ |
| | Swish | $84.84_{\pm0.69}$ | $84.20_{\pm0.24}$ | $75.68_{\pm6.62}$ | $78.38_{\pm5.67}$ | $52.87_{\pm2.57}$ |
| | **TAFS** | $\mathbf{88.47_{\pm0.50}}$ | $\mathbf{86.01_{\pm0.13}}$ | $76.22_{\pm5.19}$ | $\mathbf{79.73_{\pm3.87}}$ | $58.02_{\pm1.94}$ |
| Sage-JKNet | ReLU | $85.29_{\pm0.56}$ | $83.97_{\pm0.15}$ | $80.00_{\pm6.07}$ | $81.62_{\pm5.10}$ | $56.78_{\pm1.62}$ |
| | Tanh | $86.01_{\pm0.51}$ | $85.25_{\pm0.18}$ | $77.03_{\pm5.57}$ | $78.92_{\pm6.14}$ | $57.68_{\pm1.92}$ |
| | LeakyReLU | $85.90_{\pm0.42}$ | $85.01_{\pm0.25}$ | $80.27_{\pm6.84}$ | $81.35_{\pm4.75}$ | $57.41_{\pm2.01}$ |
| | Swish | $85.56_{\pm0.61}$ | $84.71_{\pm0.22}$ | $77.13_{\pm5.30}$ | $81.06_{\pm5.43}$ | $55.00_{\pm1.93}$ |
| | **TAFS** | $\mathbf{89.51_{\pm0.66}}$ | $\mathbf{86.73_{\pm0.20}}$ | $\mathbf{81.79_{\pm5.08}}$ | $\mathbf{82.10_{\pm5.16}}$ | $\mathbf{59.37_{\pm1.53}}$ |
| Sage-Mixhop | ReLU | $84.55_{\pm1.07}$ | $84.23_{\pm0.21}$ | $75.95_{\pm9.59}$ | $81.63_{\pm4.80}$ | $54.19_{\pm2.15}$ |
| | Tanh | $84.82_{\pm0.65}$ | $84.21_{\pm0.38}$ | $\mathbf{78.11_{\pm6.99}}$ | $81.89_{\pm5.80}$ | $53.20_{\pm1.21}$ |
| | LeakyReLU | $84.80_{\pm1.10}$ | $84.34_{\pm0.28}$ | $76.22_{\pm8.36}$ | $77.30_{\pm4.71}$ | $53.14_{\pm1.72}$ |
| | Swish | $84.18_{\pm0.55}$ | $84.69_{\pm0.30}$ | $75.95_{\pm8.15}$ | $80.27_{\pm6.74}$ | $53.20_{\pm1.86}$ |
| | **TAFS** | $\mathbf{87.77_{\pm1.40}}$ | $\mathbf{85.30_{\pm0.24}}$ | $77.77_{\pm4.39}$ | $\mathbf{83.70_{\pm4.05}}$ | $\mathbf{55.07_{\pm0.57}}$ |

