# OpenReview forum: "TAFS: Task-aware Activation Function Search for Graph Neural Networks"
_ICLR.cc/2024/Conference — Submitted to ICLR 2024_

### Official Review · Reviewer_SDpu · 2023-10-18

**Soundness:** 2 fair
**Presentation:** 4 excellent
**Contribution:** 3 good
**Rating:** 5
**Confidence:** 5

**Summary:**

The study presents a novel approach to Graph Neural Network (GNN) activation function search using bi-level optimization. An efficient algorithm is introduced that explores a search space defined by universal approximators with smoothness constraints, allowing for quick optimal function discovery. By using stochastic relaxation, the algorithm bypasses the challenges of non-differentiable objectives, outperforming existing activation functions across various GNN models and datasets, leading to a tailored GNN activation function design.

**Strengths:**

1. The idea of designing a task-aware activation function for GNNs is excellent, since it could universally enhance the performance of GNNs across different datasets and tasks.
2. The writing and expression of this paper are good.

**Weaknesses:**

1. Graph classification is crucial in graph mining due to its emphasis on global topological structures. Given its distinct objectives from node classification, it's imperative for the author to include experiments on graph classification.
2. As a universal method, TAFS should be tested on as many GNN models as possible. GCN and GraphSAGE are quite similar, while GAT and GIN are different, which should also be used as backbone to evaluate the performance of TAFS.
3. The authors should conduct a time complexity analysis and compare it with GReLU.

**Questions:**

See weaknesses.

---

> ### Author Response · Authors · 2023-11-22
> **Reply 1/2**
>
> Thank you for your appreciation of our work and for your insightful questions. We answer each below.
>
> Q1: *Graph classification is crucial in graph mining due to its emphasis on global topological structures. Given its distinct objectives from node classification, it's imperative for the author to include experiments on graph classification.*
>
> **Answer**: We add below the performance of graph classification on the ogbg-molhiv dataset. We include baselines of classical GNN as well as Top-1 ranking solution PAS+FPS. In all cases, TAFS could achieve a performance gain.
>
> **Graph level task** Dataset: ogbg-molhiv, #Graphs 41K, #Nodes per graph 25.5
> | Method  | Activation | Test ROCAUC | Param  | Time     |
> |---------|------------|-------------|--------|----------|
> | GCN     | ReLU       | 0.7606      | 0.53M  | 40Min    |
> |         | Tanh       | 0.7883      | 0.53M  | 40Min    |
> |         | Swish      | 0.7577      | 0.53M  | 40Min    |
> |         |  **TAFS**       | 0.7934      | +300   | 55Min    |
> |         |            |             |        |          |
> | GIN     | ReLU       | 0.7707      | 3.3M   | 53Min    |
> |         | Tanh       | 0.778       | 3.3M   | 53Min    |
> |         | Swish      | 0.7669      | 3.3M   | 53Min    |
> |         |  **TAFS**       | 0.7883      | +300   | 67Min    |
> |         |            |             |        |          |
> | PAS+FPs | ReLU       | 0.8169      | 27M    | 17 Hours |
> |         | Tanh       | 0.8201      | 27M    | 17 Hours |
> |         | Swish      | 0.8152      | 27M    | 17 Hours |
> |         |  **TAFS**       | 0.8203      | +0.01M | 23 Hours |
>
> Q2. *As a universal method, TAFS should be tested on as many GNN models as possible. GCN and GraphSAGE are quite similar, while GAT and GIN are different, which should also be used as backbone to evaluate the performance of TAFS.*
>
> **Answer**: We agree. Here are our detailed experiments on GAT/GIN across small and large datasets. As can be seen, TAFS achieves the best in most cases, showing the effectiveness of our method.
>
> **GAT experiments**
> |     |          |            | Cora  |      | DBLP  |      | Cornell |      | Texas |      | Chameleon |      |
> |-----|----------|------------|-------|------|-------|------|---------|------|-------|------|-----------|------|
> | GNN |          | Activation | mean  | std  | mean  | std  | mean    | std  | mean  | std  | mean      | std  |
> | GAT | Stack    | ReLU       | 84.73 | 0.65 | 85.36 | 0.13 | 55.14   | 7.57 | 55.68 | 6.3  | 28.42     | 9.75 |
> |     |          | Tanh       | 84.51 | 1.25 | 84.91 | 0.2  | 56.76   | 2.96 | 56.76 | 5.92 | 55.57     | 3.58 |
> |     |          | Swish      | 84.44 | 1.4  | 85.34 | 0.28 | 56.76   | 2.42 | 60    | 8.09 | 22.81     | 1.04 |
> |     |          | **TAFS**       | 86.14 | 1.68 | 85.91 | 0.13 | 56.59   | 3.15 | 58.92 | 7.91 | 55.97     | 3.16 |
> |     | Residual | ReLU       | 83.96 | 0.6  | 85.16 | 0.09 | 55.14   | 6.07 | 54.59 | 6.92 | 61.18     | 4.19 |
> |     |          | Tanh       | 85.62 | 0.18 | 85.25 | 0.3  | 54.59   | 5.24 | 54.05 | 2.42 | 64.65     | 2.25 |
> |     |          | Swish      | 84.44 | 0.97 | 84.49 | 0.31 | 52.43   | 5.57 | 56.76 | 8.88 | 59.96     | 1.29 |
> |     |          |  **TAFS**       | 86.47 | 1.25 | 85.99 | 0.22 | 55.11   | 7.33 | 56.81 | 8.44 | 65.66     | 3.78 |
> |     | JKNet    | ReLU       | 86.43 | 0.25 | 84.77 | 0.23 | 78.92   | 8.61 | 79.46 | 5.01 | 57.68     | 0.46 |
> |     |          | Tanh       | 86.73 | 0.92 | 85.07 | 0.13 | 75.14   | 5.24 | 78.38 | 2.96 | 58.25     | 1.01 |
> |     |          | Swish      | 85.47 | 0.8  | 84.68 | 0.2  | 82.7    | 6.07 | 77.84 | 5.24 | 54.69     | 2.36 |
> |     |          |  **TAFS**       | 87.21 | 0.47 | 85.22 | 0.18 | 82.84   | 6.26 | 79.38 | 4.52 | 64.82     | 2.01 |
> |     | Mixhop   | ReLU       | 84.25 | 1.67 | 84.1  | 0.12 | 63.78   | 7.76 | 62.7  | 7.91 | 51.32     | 2.27 |
> |     |          | Tanh       | 84.62 | 0.51 | 84.34 | 0.28 | 67.57   | 3.82 | 70.27 | 7.05 | 53.82     | 1.27 |
> |     |          | Swish      | 84.4  | 0.38 | 84.39 | 0.35 | 64.86   | 3.82 | 65.95 | 6.07 | 50.26     | 2.93 |
> |     |          |  **TAFS**       | 85.29 | 0.45 | 85.04 | 0.34 | 67.86   | 5.92 | 69.03 | 7.53 | 54.78     | 3.62 |
>
> ---
> Dataset: ogbn-proteins
> | Method    | Activation | Test ROCAUC | Param | Time    |
> |-----------|------------|-------------|-------|---------|
> | GAT       | ReLU       | 0.8176      | 2.48M | 33.6Min |
> |           | Tanh       | 0.8003      | 2.48M | 33.6Min |
> |           | Swish      | 0.8299      | 2.48M | 33.6Min |
> |           |  **TAFS**       | 0.8682      | +300  | 41.5Min |

---

> ### Author Response · Authors · 2023-11-22
> **Reply 2/2**
>
> **GIN experiments**
> |     |          |            | Cora  |      | DBLP  |      | Cornell |      | Texas |      | Chameleon |      |
> |-----|----------|------------|-------|------|-------|------|---------|------|-------|------|-----------|------|
> | GNN |          | Activation | mean  | std  | mean  | std  | mean    | std  | mean  | std  | mean      | std  |
> | GIN | Stack    | ReLU       | 84.7  | 0.18 | 84.48 | 0.27 | 52.97   | 6.3  | 69.19 | 7.37 | 31.49     | 1.76 |
> |     |          | Tanh       | 85.4  | 0.39 | 86.22 | 0.14 | 54.59   | 3.59 | 70.27 | 8.2  | 56.97     | 3.57 |
> |     |          | Swish      | 84.66 | 0.35 | 85.08 | 0.63 | 55.14   | 4.39 | 65.95 | 9.76 | 31.1      | 4.52 |
> |     |          |  **TAFS**       | 85.1  | 0.46 | 86.98 | 0.22 | 56.59   | 4.65 | 69.19 | 5.01 | 59.97     | 3.16 |
> |     | Residual | ReLU       | 84.77 | 0.48 | 84.18 | 0.21 | 55.14   | 4.39 | 72.43 | 4.65 | 30.53     | 3.07 |
> |     |          | Tanh       | 85.21 | 0.7  | 86.14 | 0.23 | 56.76   | 8.02 | 72.97 | 7.05 | 59.91     | 2.87 |
> |     |          | Swish      | 84.84 | 0.5  | 84.51 | 0.32 | 55.68   | 4.39 | 74.59 | 6.53 | 31.36     | 1.47 |
> |     |          |  **TAFS**       | 85.06 | 0.51 | 88.27 | 0.23 | 59.59   | 3.97 | 77.69 | 6.07 | 60.66     | 3.97 |
> |     | JKNet    | ReLU       | 85.95 | 0.78 | 84.96 | 0.23 | 71.89   | 6.53 | 82.7  | 6.07 | 31.45     | 1.8  |
> |     |          | Tanh       | 85.4  | 0.65 | 85.68 | 0.27 | 72.97   | 4.83 | 82.16 | 3.67 | 64.47     | 1.71 |
> |     |          | Swish      | 84.99 | 0.44 | 85.04 | 0.24 | 70.27   | 6.16 | 82.16 | 6.07 | 33.16     | 3.25 |
> |     |          |  **TAFS**       | 85.8  | 0.32 | 86.61 | 0.12 | 74.59   | 5.57 | 83.62 | 6.26 | 66.71     | 3.44 |
> |     | Mixhop   | ReLU       | 83.66 | 0.92 | 81.69 | 0.13 | 65.41   | 3.59 | 77.84 | 7.13 | 56.14     | 3.07 |
> |     |          | Tanh       | 83.59 | 0.84 | 81.78 | 0.2  | 65.41   | 3.59 | 72.97 | 9.67 | 56.18     | 2.19 |
> |     |          | Swish      | 83.22 | 1.3  | 81.59 | 0.54 | 66.49   | 3.24 | 76.76 | 7.76 | 56.4      | 1.75 |
> |     |          |  **TAFS**       | 83.96 | 0.46 | 81.83 | 0.2  | 71.35   | 6.96 | 79.84 | 6.71 | 64.95     | 2.43 |
>
> ---
> Dataset: ogbg-molhiv
> | Method | Activation | Test ROCAUC | Param | Time  |
> |--------|------------|-------------|-------|-------|
> | GIN    | ReLU       | 0.7707      | 3.3M  | 53Min |
> |        | Tanh       | 0.778       | 3.3M  | 53Min |
> |        | Swish      | 0.7669      | 3.3M  | 53Min |
> |        |  **TAFS**       | 0.7883      | +300  | 67Min |
>
> Q3. * The authors should conduct a time complexity analysis and compare it with GReLU.*
>
> **Answer**: The setting of GReLU is actually very different from ours. GReLU designs a hyperfunction which includes additional graph convolutional layers, as a result, **GReLU is not a univariate function** as commonly required for activation functions (on the other hand, ReLU, Swish, Tanh and our TAFS as univariate functions). This can also be perceived by GReLU’s parameters. GReLU has N * C * K extra parameters, where N is number of nodes, C is the node feature dimensions and K is number of segments. So **GReLU is dependent on the graph dataset size**. TAFS is independent. Our TAFS brings extra parameter only dependent to number of graph layers, which is basically M * L, where M is the size of MLP (usually dozens) and L is the applied layers.

---

### Official Review · Reviewer_vEQt · 2023-10-31

**Soundness:** 3 good
**Presentation:** 3 good
**Contribution:** 2 fair
**Rating:** 5
**Confidence:** 4

**Summary:**

This paper proposes a Task-Aware Activation Function Search method, abbreviated as TAFS. TAFS is capable of efficiently searching for and discovering new, effective activation functions within GNN applications. Firstly, for the search space of activation functions, TAFS introduces a continuous latent space equipped with a general approximator that includes an additional smoothness constraint. Specifically, TAFS utilizes a MLP to approximate the optimal activation function, where the parameters of the MLP are optimized as part of the search space, incorporating a Jacobian regularization term. Secondly, employing stochastic relaxation techniques, the search space is reparameterized with a Gaussian distribution, shifting the optimization target from the parameters of the activation function to those of the Gaussian distribution. Comprehensive evaluations on node and link-level tasks demonstrate that this method achieves excellent performance.

**Strengths:**

1. Presents an innovative and intriguing approach to activation function search within the context of Graph Neural Networks, marking a novel area of research.
2. Introduces a probabilistic search algorithm capable of effectively exploring a regularized function space, leading to the discovery of novel activation functions.
3. Experimental results demonstrate that, compared to baseline methods, TAFS achieves excellent performance in node and link prediction tasks, significantly enhancing the efficiency of the search process.

**Weaknesses:**

1. The search strategy presented in this paper does not support a larger GNN search space. For instance, if there is a need to search for aggregation functions or message passing functions, the optimal form of the activation function is likely to change.

2. If TAFS employs an MLP to approximate the optimal activation function, the performance of activation function would also depend on the number of layers in the MLP and the non-linear transformation functions. This has not been discussed in this paper.

3. In experiments, although the paper introduces a Jacobian regularization term, the impact of this regularization has not been empirically tested.

4. In TAFS, a stochastic relaxation is used, involving the reparameterization of the activation function parameters using a Gaussian distribution. However, this paper does not discuss the advantage of this strategy.

**Questions:**

1. In a previous work [1], the best-performing activation function is referred to as Swish, which has a specific functional form. Similarly, can the best-performing activation function identified by TAFS be represented using a generic function?
[1] Prajit Ramachandran, Barret Zoph, and Quoc V . Le. Searching for activation functions. In International Conference on Learning Representations, ICLR, 2018.

2. The sentence in the main text, "… how can we design GNN activation functions to adapt effectively to various graph-based tasks, creating task-aware activation functions??" seems to have an issue with the use of symbols.

3. In the text, "... and demote by ¯w_δ all the parameters of GNN, …", the word "demote" appears to be misspelled.

4. The experimental section lacks clear explanations of the metrics used in the tables. For example, the description of Table 2 does not specify whether the results represent accuracy, AUC, or something else.

5. In Figure 4, subfigure (a) shows that larger values of K yield better results but at a slower pace, demonstrating a "trade-off between accuracy and computation time." Is the Y-axis in (a) representing accuracy? And it's not clear how the relationship between K value and computation time is depicted.

---

> ### Author Response · Authors · 2023-11-22
> **Reply 1/2**
>
> Thank the reviewer for insightful questions. We answer each below.
>
> Q1. *The search strategy presented in this paper does not support a larger GNN search space. For instance, if there is a need to search for aggregation functions or message passing functions, the optimal form of the activation function is likely to change.*
>
> **Answer**: Actually, this is totally integratable. Our method TAFS could be applied after the other GNN search method finds the backbone model (for example, GraphNAS, 2019 Gao et al). We experiment this strategy on the OGB graph classification top 1 winner PAS+FPS method. It is a neural architecture search baseline that does exactly GNN space search. In the searched 14 layers GNN checkpoint, we replace its ReLU functions by TAFS, and fine tune further. As can be seen, within an affordable time, the performance could boost further, showing that TAFS is easily integrable with the GNN NAS methods.
>
> | Method  | Activation | Test ROCAUC | Param  | Time     |
> |---------|------------|-------------|--------|----------|
> | PAS+FPs | ReLU       | 0.8169      | 27M    | 17 Hours |
> |         | TAFS       | 0.8203      | +0.01M | 23 Hours |
>
> Reference:  https://ogb.stanford.edu/docs/leader_graphprop/#ogbg-molhiv
>
> Q2. *If TAFS employs an MLP to approximate the optimal activation function, the performance of activation function would also depend on the number of layers in the MLP and the non-linear transformation functions. This has not been discussed in this paper.*
>
> **Answer**:  **We highlight that MLP is a universal approximator which we choose on purpose as an expressive functional space. As a result, a normal MLP (not too large) would be sufficient in finding the suitable functions.**
>
> We include the discussion on MLP hyperparameters in Figure 4(b)(c). It is true that this influences the performance. In order not to add too many extra parameters, we choose MLP with one-hidden layer (100 neurons) of Tanh activation function. No activation function is applied on the output neuron. We experiment below the differences of ReLU/Tanh used in MLP. Tanh is thus our final choice.
>
> Dataset: DBLP
>
> | Model        | Activation  | Accuracy |
> |--------------|-------------|----------|
> | GCN-Stack    | ReLU        | 83.06    |
> |              | TAFS (relu) | 88.91    |
> |              | TAFS (tanh) | 89.08    |
> | GAT-residual | ReLU        | 83.96    |
> |              | TAFS (relu) | 85.23    |
> |              | TAFS (tanh) | 85.47    |
>
> Q3. *In experiments, although the paper introduces a Jacobian regularization term, the impact of this regularization has not been empirically tested.*
>
> **Answer**:  Thanks for pointing this out. We give below the influence of TAFS w./w.o. regularization. We will add this part to the paper later.
>
> Dataset: DBLP
> | Model        | Activation      | Accuracy |
> |--------------|-----------------|----------|
> | GCN-Stack    | ReLU            | 83.06    |
> |              | TAFS (w.o. reg) | 87.51    |
> |              | TAFS (w. Reg)   | 89.08    |
> | GAT-residual | ReLU            | 83.96    |
> |              | TAFS (w.o. reg) | 83.23    |
> |              | TAFS (w. Reg)   | 85.47    |
>
> Q4. *In TAFS, a stochastic relaxation is used, involving the reparameterization of the activation function parameters using a Gaussian distribution. However, this paper does not discuss the advantage of this strategy.*
>
> **Answer**: The biggest advantage of the stochastic relaxation is to remove the differentiability requirement for the task evaluation metric (e.g. ROCAUC, Hit@20), which is very common in Graph tasks. By sampling the weights of activation functions from a probability distribution, we optimize the distribution parameters by proposition 1 in the paper. The proposition 1 gives the gradient of loss w.r.t distribution parameters, and there is no terms such as \nable M, avoiding the calculation of gradient of M. This makes the method more applicable in general tasks. The choice of Gaussian distribution is without loss of generality because we don’t have prior knowledge about this choice of distribution. In cases where we know certain probability distribution is better, we could definitely change to others such Poisson distribution.
>
> We give below the influence of TAFS w./w.o. stochastic regularization. The full table has been updated in Table 3 in the paper.
> | Dataset | Model   | Activation                    | ROCAUC | PRAUC |
> |---------|---------|-------------------------------|--------|-------|
> | DTI     | SkipGNN | ReLU                          | 0.922  | 0.928 |
> |         |         |        TAFS w.o. relaxation   | 0.933  | 0.934 |
> |         |         | TAFS                          | 0.952  | 0.954 |
> |         | HOGCN   | ReLU                          | 0.927  | 0.929 |
> |         |         |        TAFS w.o. relaxation   | 0.923  | 0.922 |
> |         |         | TAFS                          | 0.943  | 0.94  |

---

> ### Author Response · Authors · 2023-11-22
> **Reply 2/2**
>
> Q5. *In a previous work [1], the best-performing activation function is referred to as Swish, which has a specific functional form. Similarly, can the best-performing activation function identified by TAFS be represented using a generic function? [1] Prajit Ramachandran, Barret Zoph, and Quoc V . Le. Searching for activation functions. In International Conference on Learning Representations, ICLR, 2018.*
>
> **Answer**: This is an interesting question. We answer in two ways. Firstly, we could achieve the closed form equation by symbolic regression techniques. Because we have obtained the univariate function and we could simulate many (x, y) value pairs to symbolic regression to distill. Swish could do so because it uses a template based space which naturally leads to this symbolic form as a result. However, this has a big negative impact on the possible function space, i.e., it is very possible the suitable activation function is not in this manually-designed template space. Secondly, in terms of usage, we do not need to distill the explicit form because we open source the learned weights of the univariate function and for practitioners, they just need to load this small function and replace their ReLU. The usage is thus very easy.
>
> We show below a case study of the symbolized version of TAFS searched results. On the Cora dataset with GCN-JK network baseline, the searched result is the blue line. We distill an explicit formula by symbolic regression: y = 0.6 Tanh(-x) and plug in this activation function back to the model. The performance is shown below and we provide a **visualization in the appendix Sec H**. There is a performance gap in the table below. This is because explicit symbolic space is not accurate, which shows further that our implicit functional space is expressive.
>
> | Dataset | Baseline | Activation   | Accuracy |
> |---------|----------|--------------|----------|
> | Cora    | GCN-JK   | TAFS         | 89.08    |
> |         |          | 0.6*Tanh(-x) | 87.89    |
>
> Q6-Q7: *The sentence in the main text, "… how can we design GNN activation functions to adapt effectively to various graph-based tasks, creating task-aware activation functions??" seems to have an issue with the use of symbols.*
>
> *In the text, "... and demote by ¯w_δ all the parameters of GNN, …", the word "demote" appears to be misspelled.*
>
> **Answer**: Thanks for your careful review. We have corrected these typos.
>
> Q8: * The experimental section lacks clear explanations of the metrics used in the tables. For example, the description of Table 2 does not specify whether the results represent accuracy, AUC, or something else.*
>
> **Answer**:  In Table 2, the standard accuracy is used. In Table 3, both ROCAUC and PRAUC are used. In our extended OGB experiments, we use  ROCAUC for node classification and graph classification, Hit@20 for link prediction, following the OGB standard.Thus, we have experimented on various tasks of different objective requirements.
>
> Q9: *In Figure 4, subfigure (a) shows that larger values of K yield better results but at a slower pace, demonstrating a "trade-off between accuracy and computation time." Is the Y-axis in (a) representing accuracy? And it's not clear how the relationship between K value and computation time is depicted.*
>
> **Answer**: Yes, Y-axis is the downstream performance, here classification accuracy. For example, moving from K=32 towards K=64 increases 20% more computational times. As a result, it is truly a trade-off that should depend on the user's own preference. We will add this discussion in the paper.

---

### Official Review · Reviewer_kgxw · 2023-10-31

**Soundness:** 3 good
**Presentation:** 3 good
**Contribution:** 2 fair
**Rating:** 5
**Confidence:** 3

**Summary:**

The authors introduce a framework for designing activation functions for graph neural networks (GNNs) based on the downstream task. The framework, called TAFS, uses a bi-level stochastic optimization problem with Lipschitz regularization to search for the optimal activation patterns. The authors claim that TAFS can automate the discovery of task-aware activation functions without significant computational or memory overhead and show that TAFS can achieve substantial improvements over existing methods in various tasks.

**Strengths:**

S1. The authors review the existing work, explain the necessity of achieving task-aware activation functions in the context of GNNs, and identify two challenges for this goal. To address them, the authors design a new framework consisting of a compact search space and an efficient search algorithm, which enables automated activation function search.

S2. The paper writing is relatively good, and the authors provide a detailed explanation of the key parts of their method, namely the implicit functional search space and the stochastic relaxation.

**Weaknesses:**

W1. Some of the experimental settings and results explanations in the paper are vague, such as Figure 1 in the Introduction section and Figure 3 in the Experiments section. I cannot understand how the experimental results in the figures were obtained, and what the data in the figures mean.

W2. From the experimental results, the performance of TAFS needs to be improved. Table 2 shows that in some experimental scenarios on the DBLP, Cornell, Texas and Chameleon datasets, the results obtained by TAFS are only marginally better than directly using a specific activation function, or even worse. The authors did not explain this phenomenon.

W3. I wonder why in the drug and protein interaction prediction experiments, only the results of TAFS and ReLU were provided, instead of comparing with multiple activation functions as in the node classification experiments.

W4. Since the authors compared the search efficiency with Swish and APL, why didn’t they involve the comparison with these two search-based methods in the effectiveness validation experiments (Table 2 and Table 3)?

**Questions:**

See the weakness part

---

> ### Author Response · Authors · 2023-11-22
> **Reply 1/2**
>
> Thanks for the questions. We answer each below.
>
> Q1. *Some of the experimental settings and results explanations in the paper are vague, such as Figure 1 in the Introduction section and Figure 3 in the Experiments section. I cannot understand how the experimental results in the figures were obtained, and what the data in the figures mean.*
>
> **Answer**: In a word, Figure 1 illustrates the impact of activation function in GNN and motivates our work to search activation functions. Figure 3 demonstrates the searched candidates from baselines and our proposed TAFS.
>
> **Figure 1** shows the experiments on two GNN baselines (GCN and GraphSage) on two typical graph datasets (Core and DBLP), with different choices of activation functions (ReLU, Tanh, Leaky-Relu, Swish). The evaluation metric is node classification accuracy. We show by Figure 1 that activation functions in GNN, which was previously paid little attention, could have a huge impact on the performance. This motivates our study to design efficient activation function search algorithms in this paper.
>
> **Figure 3** demonstrates multiple things. Fig 3(a) gives an idea on the search space of our compared baselines Swish and APL to show what functions in their search space looks like. Fig 3(b)(c) shows our searched activation functions tailoring two datasets Cora and DBLP. Specifically, we show that (1) our method finds adaptive activation functions for different datasets (2) due to the efficiency of our method, we could adapt layer-wise activation functions without exploding the search space. So different layers find different activation functions (3) our found activation functions are distinct to already published manually-designed ones (4) our activation functions are smooth functions which potentially reflects our smoothness regularization.
>
> Q2. *From the experimental results, the performance of TAFS needs to be improved. Table 2 shows that in some experimental scenarios on the DBLP, Cornell, Texas and Chameleon datasets, the results obtained by TAFS are only marginally better than directly using a specific activation function, or even worse. The authors did not explain this phenomenon.*
>
> **Answer**: Thank the reviewer for pointing this out. In our previously table version, we bold the best results by considering only the mean over multiple runs. Actually, to make it more precise, we now take into account both the mean and standard variance and bold best results of the same statistical significance level.
>
> In the revised pdf, we see now that TAFS is best or tied for best in all cases. In some datasets which are small such as Texas, Chameleon, the variance is very high. Thus, TAFS archives the same performance level.
>
> This phenomenon is much alleviated in large scale datasets such as OGB ones. In our experiments with OGB, we show that TAFS is the best as below.
>
> **Node level task** Dataset: ogbn-proteins, Node 132,534, Edge 39,561,252
> | Method    | Activation | Test ROCAUC | Param | Time       |
> |-----------|------------|-------------|-------|------------|
> | GCN       | ReLU       | 0.7219      | 100K  | 15.3Min    |
> |           | Tanh       | 0.6521      | 100K  | 15.3Min    |
> |           | Swish      | 0.7255      | 100K  | 15.3Min    |
> |           | **TAFS**       | **0.7483**      | +300  | 16.8Min    |
> | GAT       | ReLU       | 0.8176      | 2.48M | 33.6Min    |
> |           | Tanh       | 0.8003      | 2.48M | 33.6Min    |
> |           | Swish      | 0.8299      | 2.48M | 33.6Min    |
> |           | **TAFS**       | **0.8682**      | +300  | 41.5Min    |
> | GIPA      | ReLU       | 0.8917      | 17M   | 10.7 hours |
> |           | Tanh       | 0.8809      | 17M   | 10.7 hours |
> |           | Swish      | 0.8899      | 17M   | 10.7 hours |
> |           | **TAFS**       | **0.8991**      | +1500 | 14.9 hours |
>
> ---
> **Link level task** dataset: ogbl-ddi, Node 4,267, Edge 1,334,889
> | Method    | Activation | Test Hits@20 | Param | Time      |
> |-----------|------------|--------------|-------|-----------|
> | GCN       | ReLU       | 0.3707       | 1.29M | 7Min      |
> |           | Tanh       | 0.3318       | 1.29M | 7Min      |
> |           | Swish      | 0.3772       | 1.29M | 7Min      |
> |           | **TAFS**       |**0.3991**      | +300  | 10Min     |
> | GraphSage | ReLU       | 0.5391       | 1.42M | 8Min      |
> |           | Tanh       | 0.5228       | 1.42M | 8Min      |
> |           | Swish      | 0.5377       | 1.42M | 8Min      |
> |           | **TAFS**       | **0.5512**       | +300  | 13Min     |
> | E2N       | ReLU       | 0.9606       | 0.6M  | 4 Hours   |
> |           | Tanh       | 0.9554       | 0.6M  | 4 Hours   |
> |           | Swish      | 0.9515       | 0.6M  | 4 Hours   |
> |           | **TAFS**       | **0.9681**       | +300  | 4.7 Hours |

---

> ### Author Response · Authors · 2023-11-22
> **Reply 2/2**
>
> Q3. *I wonder why in the drug and protein interaction prediction experiments, only the results of TAFS and ReLU were provided, instead of comparing with multiple activation functions as in the node classification experiments.*
>
> **Answer**: We add the other choices’ performance below. We didn’t include this to emphasize the impact by adopting TAFS.  The table will be merged to the main paper later.
>
> | Dataset | Model   | Activation | ROCAUC | PRAUC |
> |---------|---------|------------|--------|-------|
> | DTI     | SkipGNN | ReLU       | 0.922  | 0.928 |
> |         |         | Tanh       | 0.915  | 0.911 |
> |         |         | Swish      | 0.931  | 0.935 |
> |         |         | **TAFS**       | 0.952  | 0.954 |
> |         | HOGCN   | ReLU       | 0.927  | 0.929 |
> |         |         | Tanh       | 0.903  | 0.911 |
> |         |         | Swish      | 0.925  | 0.921 |
> |         |         | **TAFS**       | 0.943  | 0.94  |
> |         |         |            |        |       |
> | DDI     | SkipGNN | ReLU       | 0.886  | 0.866 |
> |         |         | Tanh       | 0.903  | 0.872 |
> |         |         | Swish      | 0.881  | 0.865 |
> |         |         | **TAFS**       | 0.911  | 0.898 |
> |         | HOGCN   | ReLU       | 0.898  | 0.881 |
> |         |         | Tanh       | 0.887  | 0.876 |
> |         |         | Swish      | 0.881  | 0.872 |
> |         |         | **TAFS**       | 0.917  | 0.901 |
> |         |         |            |        |       |
> | PPI     | SkipGNN | ReLU       | 0.917  | 0.921 |
> |         |         | Tanh       | 0.912  | 0.919 |
> |         |         | Swish      | 0.92   | 0.921 |
> |         |         | **TAFS**       | 0.927  | 0.937 |
> |         | HOGCN   | ReLU       | 0.919  | 0.922 |
> |         |         | Tanh       | 0.917  | 0.913 |
> |         |         | Swish      | 0.915  | 0.914 |
> |         |         | **TAFS**       | 0.923  | 0.929 |
> |         |         |            |        |       |
> | DGA     | SkipGNN | ReLU       | 0.912  | 0.915 |
> |         |         | Tanh       | 0.912  | 0.914 |
> |         |         | Swish      | 0.924  | 0.917 |
> |         |         | **TAFS**       | 0.93   | 0.94  |
> |         | HOGCN   | ReLU       | 0.927  | 0.934 |
> |         |         | Tanh       | 0.918  | 0.929 |
> |         |         | Swish      | 0.923  | 0.924 |
> |         |         | **TAFS**       | 0.933  | 0.942 |
>
> Q4. *Since the authors compared the search efficiency with Swish and APL, why didn’t they involve the comparison with these two search-based methods in the effectiveness validation experiments (Table 2 and Table 3)?*
>
> **Answer**: Due to the inefficiency of Swish and APL, we could not practically obtain reasonable results by Searchable Swish and APL. For example, in the case of Table 4 in the paper, we can see that searchable Swish takes 50 Hours to run on Chameleon and APL costs 6 times more memory consumption than TAFS. In our table 2, we have included 5 datasets and 10 graph baselines, leading to a total of 50 sets of experiments. We find it not necessary to include the searchable Swish and APL in every single set of task. As a result, we included already in Table 2 and Table 3 the searched results of Swish paper (Swish), which is worse than TAFS. We also give below the APL results and searchable swish on two tasks for a general idea of their ineffectiveness. Note that both baselines could not be applied in OGB datasets due to inefficiency in time and memory, while TAFS could.
>
> Dataset: DBLP
>
> | Model        | Activation   | Accuracy |
> |--------------|--------------|----------|
> | GCN-Stack    | ReLU         | 83.06    |
> |              | Swish search | 82.79    |
> |              | APL search   | 81.03    |
> |              | **TAFS**         | **89.08**    |
> | GAT-residual | ReLU         | 83.96    |
> |              | Swish search | 83.01    |
> |              | APL search   | 81.22    |
> |              | **TAFS**         | **85.47**    |

---

### Official Review · Reviewer_JLTs · 2023-11-01

**Soundness:** 2 fair
**Presentation:** 3 good
**Contribution:** 1 poor
**Rating:** 5
**Confidence:** 3

**Summary:**

The paper introduces TAFS, a framework for designing task-specific activation functions in Graph Neural Networks (GNNs). TAFS uses a search algorithm to optimize activation functions for specific tasks, resulting in improved performance compared to traditional activation functions. The design of TAFS is more efficient than baseline methods on optimizing such bi-level optimization problems. The experiment results show that using TAFS usually achieves better performances than using fixed activation functions.

**Strengths:**

1. The paper is overall clearly written and easy to follow.
2. This paper studies a interesting research problem that is rarely studied in the GML domain.

**Weaknesses:**

1. The experimental results are not very convincing. The experiments are all conducted on very small datasets, and I don't think the datasets for link prediction experiments are the commonly used ones in literature. I'd recommend the authors to use more standardized and commonly accepted benchmarks such as the OGB ones.
2. Although the proposed method is already much more efficient than the baselines Swish and APL, it still takes about 10x times of runtime when compared with fixed activation functions. It's hard to tell whether the performance improvements worth such huge overhead on the time cost.

**Questions:**

1. I'd appreciate if the authors can elaborate more on the difference of the proposed method versus the activation function search methods for CNNs and RNNs (as referenced in Sec. 1).

---

> ### Author Response · Authors · 2023-11-22
> **Reply 1/2**
>
> Thanks for the important suggestion on the scalability. We answer each question below.
>
> Q1: *The experimental results are not very convincing. The experiments are all conducted on very small datasets, and I don't think the datasets for link prediction experiments are the commonly used ones in literature. I'd recommend the authors to use more standardized and commonly accepted benchmarks such as the OGB ones.
> Although the proposed method is already much more efficient than the baselines Swish and APL, it still takes about 10x times of runtime when compared with fixed activation functions. It's hard to tell whether the performance improvements worth such huge overhead on the time cost.*
>
> Q2. *Although the proposed method is already much more efficient than the baselines Swish and APL, it still takes about 10x times of runtime when compared with fixed activation functions. It's hard to tell whether the performance improvements worth such huge overhead on the time cost.*
>
> **Answer**: We answer Q1 and Q2 together. The inappropriate comparison in the manuscript was due to the fact that the baseline was too small and as a result the computational head seems to be 10x more, though our method is already much more efficient than literature. Actually, **our method’s parameters have no dependency on the size of the graph**, making it easily scalable. We experimented on **OGB and much larger baselines** as in the table below. Three OGB datasets in node/link/graph level tasks are chosen and we experiment both classical GNN baselines and top-ranking baselines. As can be seen, on larger datasets (ogbn-molhiv) and larger baselines (PAS) which take over 17 hours, our method only adds about 35% more computational time, which is much more acceptable than baselines. Actually, both **Swish search and APL search could not be run on such larger datasets due to their inefficiency** in running time and overparameterization, which leads to months’ GPU hours or leads to Out-of-Memory error.
>
> ---
> **Node level task** Dataset: ogbn-proteins, Node 132,534, Edge 39,561,252
>
> | Method    | Activation | Test ROCAUC | Param | Time       |
> |-----------|------------|-------------|-------|------------|
> | GCN       | ReLU       | 0.7219      | 100K  | 15.3Min    |
> |           | Tanh       | 0.6521      | 100K  | 15.3Min    |
> |           | Swish      | 0.7255      | 100K  | 15.3Min    |
> |           | TAFS       | 0.7483      | +300  | 16.8Min    |
> |           |            |             |       |            |
> | GraphSage | ReLU       | 0.7567      | 193K  | 17.7Min    |
> |           | Tanh       | 0.7442      | 193K  | 17.7Min    |
> |           | Swish      | 0.7319      | 193K  | 17.7Min    |
> |           | TAFS       | 0.7768      | +300  | 19.9Min    |
> |           |            |             |       |            |
> | GAT       | ReLU       | 0.8176      | 2.48M | 33.6Min    |
> |           | Tanh       | 0.8003      | 2.48M | 33.6Min    |
> |           | Swish      | 0.8299      | 2.48M | 33.6Min    |
> |           | TAFS       | 0.8682      | +300  | 41.5Min    |
> |           |            |             |       |            |
> | GIPA      | ReLU       | 0.8917      | 17M   | 10.7 hours |
> |           | Tanh       | 0.8809      | 17M   | 10.7 hours |
> |           | Swish      | 0.8899      | 17M   | 10.7 hours |
> |           | TAFS       | 0.8991      | +1500 | 14.9 hours |
>
> ---
> **Link level task** dataset: ogbl-ddi, Node 4,267, Edge 1,334,889
>
> | Method    | Activation | Test Hits@20 | Param | Time      |
> |-----------|------------|--------------|-------|-----------|
> | GCN       | ReLU       | 0.3707       | 1.29M | 7Min      |
> |           | Tanh       | 0.3318       | 1.29M | 7Min      |
> |           | Swish      | 0.3772       | 1.29M | 7Min      |
> |           | TAFS       | 0.3991       | +300  | 10Min     |
> |           |            |              |       |           |
> | GraphSage | ReLU       | 0.5391       | 1.42M | 8Min      |
> |           | Tanh       | 0.5228       | 1.42M | 8Min      |
> |           | Swish      | 0.5377       | 1.42M | 8Min      |
> |           | TAFS       | 0.5512       | +300  | 13Min     |
> |           |            |              |       |           |
> | E2N       | ReLU       | 0.9606       | 0.6M  | 4 Hours   |
> |           | Tanh       | 0.9554       | 0.6M  | 4 Hours   |
> |           | Swish      | 0.9515       | 0.6M  | 4 Hours   |
> |           | TAFS       | 0.9681       | +300  | 4.7 Hours |

---

> ### Author Response · Authors · 2023-11-22
> **Reply 2/2**
>
> **Graph level task** Dataset: ogbg-molhiv, #Graphs 41K, #Nodes per graph 25.5
>
> | Method  | Activation | Test ROCAUC | Param  | Time     |
> |---------|------------|-------------|--------|----------|
> | GCN     | ReLU       | 0.7606      | 0.53M  | 40Min    |
> |         | Tanh       | 0.7883      | 0.53M  | 40Min    |
> |         | Swish      | 0.7577      | 0.53M  | 40Min    |
> |         | TAFS       | 0.7934      | +300   | 55Min    |
> |         |            |             |        |          |
> | GIN     | ReLU       | 0.7707      | 3.3M   | 53Min    |
> |         | Tanh       | 0.778       | 3.3M   | 53Min    |
> |         | Swish      | 0.7669      | 3.3M   | 53Min    |
> |         | TAFS       | 0.7883      | +300   | 67Min    |
> |         |            |             |        |          |
> | PAS+FPs | ReLU       | 0.8169      | 27M    | 17 Hours |
> |         | Tanh       | 0.8201      | 27M    | 17 Hours |
> |         | Swish      | 0.8152      | 27M    | 17 Hours |
> |         | TAFS       | 0.8203      | +0.01M | 23 Hours |
>
> ---
>
> Q3. *I'd appreciate if the authors can elaborate more on the difference of the proposed method versus the activation function search methods for CNNs and RNNs (as referenced in Sec. 1).*
>
> **Answer**: The two most important activation function search methods are Swish and APL, which is explained and compared in Sec 2 and especially in Table 1. We give a detailed introduction in Appendix B. Swish is inefficient in running time because every inner optimization will train the network until convergence. APL is inefficient in parameters and memory because it parameterized each neuron’s activation by multiple hinges. In the case of ogbg-molhiv datasets (above) and 1st ranking solution PAS on graph classification, the total parameter is 27M and the running time is over 17 hours. APL will add over 150M parameters which leads to OOM and Swish will cost over 70 days even if we train only 100 epochs. Our method TAFS, takes only 23 hours and adds only 0.01M parameters, which is significantly more efficient than baselines, and as a result, paves the way for further study in activation function search.
>
> | Dataset     | Model   | Param  | Time     |
> |-------------|---------|--------|----------|
> | ogbg-molhiv | PAS+FPS | 27M    | 17Hours  |
> |             | Swish   | NA     | >70days  |
> |             | APL     | OOM (>150M)   | NA       |
> |             | TAFS    | +0.01M | 23 Hours |

---

> > ### Comment · Reviewer_JLTs · 2023-12-01
> >
> > Thanks for the rebuttal. I raised my score to 5.

---

### Author Response · Authors · 2023-11-22
**General response**

We thank all the reviewers for the insightful comments. Our work has been strengthened with **significantly more experiments and analysis.** We highlight motivation and contributions here and reply to each question separately later

**Motivation**: Activation function is important for Graph Neural Networks (GNN). GNN usually has shallow layers. GNN is just a simple transformation of node features without activation functions. Activation functions empower GNN to increase the nonlinear modeling capacity [1][2]. Compared to CNN which is usually deep, GNN can have larger dependency on the appropriate choice of activation functions.

This motivates us to explore activation function searching, which is paid little attention in the literature.

Our contributions are multiple-folds:
1. **Novelty**: We achieve task-awareness by stochastic relaxation and the proposed MLP gives expressive and compact parameterization for implicit functional space.
    - compared to **Swish**: Swish takes much more time because it requires the inner optimization to be converged while TAFS is iterated bi-level optimization which does not requires inner convergence at each iteration
    - compared to **APL**: APL takes 6 times more parameters that cannot be applied in large graph while TAFS is scalable
    - compared to **GReLU**: GReLU is not univariate and has a number of parameters dependent on the graph size, while TAFS’ parameter is independent of graph size. Also, GReLU is not searching algorithm
2. **Effectiveness**: We experiment on both small and large datasets. We could achieve consistent improvement w.r.t ReLU choice and on diverse tasks in node/link/graph level, our method is the state-of-the-art in activation function design.
3. **Efficiency**: We experiment on large OGB dataset and top ranking solutions up to 27M parameters. We show that TAFS finds appropriate solutions with extra 35% computational time and negligible extra memory cost, which is not possible for Swish or APL search methods.

As a consequence, our solution provides significant value for practitioners and to the community. Our proposed TAFS is an off-the-shelf tool for better activation function design, which provides a more efficient and effective baseline for pushing further research, emphasizing the previously neglected design part of GNN.

1. Prajit Ramachandran, Barret Zoph, Quoc V. Le: Searching for Activation Functions. ICLR  2018
2. Bianca Iancu, Luana Ruiz, Alejandro Ribeiro, Elvin Isufi: Graph-Adaptive Activation Functions for Graph Neural Networks. MLSP 2020

---

### Meta-Review · Area_Chair_HSgy · 2023-12-06

**Metareview:**

TAFS is an interesting method which puts activation functions in the spotlight of GNN design, offering a novel method for automated optimisation of activation functions in a bi-level setting.

I find the approach to be interesting, the topic timely, and the results have potential for a strong paper -- especially taking into account the rebuttal results, which the authors sadly haven't incorporated into the main paper during the discussion phase.

In my opinion, there are still several areas of improvement for the work as currently presented (including the rebuttal), but to keep this meta-review concise, I highlight the most actionable one -- the presentation and discussion of the OGB results.

The authors added _a lot_ of OGB results during the rebuttal, however they provided no error bars / standard deviations around their estimates -- which, in absence of any other data, forces me to conclude that these are single-run results. Further, there are no details provided about the models considered in the OGB tables; most importantly: whether the gains in performance could have also been obtained through careful hyperparameter tuning outside of the activation function. And I am doubtful that ReLU is the only useful baseline to use for these tasks---there are other "fixed" activation functions that could have been compared against.

I would have much rather preferred if the authors had focussed only on one OGB task/base architecture, but invested more effort into robust error estimates and actually convincing us that the stated gains came from TAFS and could not have been easily obtained in other ways. In the present form, especially given unanimous rejection ratings by all reviewers, I think a rejection is the most sensible outcome for this work. I do hope the authors will revise, refine and contextualise their OGB results, and come back stronger in the next revision cycle.

**Justification For Why Not Higher Score:**

Reviewers unanimously recommend rejection after rebuttal, and more work is needed to robustify the results presented within it.

**Justification For Why Not Lower Score:**

N/A

---

### Decision · Program_Chairs · 2024-01-16

Reject